# Bridging nanomechanics and bioenergetics of single mitochondria by atomic force microscopy

Ekaterina O Zorikova[1], Sabita Chourasia[1], Irit Rosenhek-Goldian[2], Sidney R Cohen[2], Semen V Nesterov[3], Atan Gross[1]

Mitochondria orchestrate energy conversion and cell fate, yet label-free approaches that report both functional and physical states at the single-organelle level are nonexistent. Here, we combine atomic force microscopy (AFM) imaging with single-mitochondrion phenotyping by quantifying stiffness, height, and spontaneous low-frequency height fluctuations at the nanoscale. Across respiratory activators, inhibitors, and uncouplers, the integrated 0- to 20-Hz fluctuation power correlates with mitochondrial membrane potential ($\Delta\Psi_m$) and does not covary with changes in mitochondrial height (a proxy for swelling). In liver mitochondria lacking mitochondrial carrier homolog 2 (MTCH2), a regulator of mitochondrial metabolism, dynamics, and apoptosis, AFM reveals a compact, mechanically stiff, high-fluctuation state consistent with hyperpolarization and distinct from inhibited/uncoupled signatures. Extending the assay to mitochondria isolated from mouse embryonic fibroblasts, AFM data can distinguish between genotypes: loss of the mitochondrial pro-fusion proteins mitofusin 1 or 2 (*MFN1* or *MFN2*) yields stiff, low-fluctuation mitochondria with reduced $\Delta\Psi_m$, whereas MTCH2 loss produces stiff, high-fluctuation, high-$\Delta\Psi_m$ mitochondria. These three label-free features provide reproducible single-organelle "fingerprints" that resolve bioenergetic states and molecular defects and complement fluorescence and respirometry.

## Introduction

Mitochondria coordinate cellular metabolism and energy production by generating ATP via oxidative phosphorylation (OXPHOS). The electron transport chain, comprising complexes I–IV together with ATP synthase (complex V), resides in the inner mitochondrial membrane and establishes the proton-motive force whose electrical component, the mitochondrial membrane potential ($\Delta\Psi_m$), drives ATP synthesis (1). In addition to their role in bioenergetics, mitochondria regulate signaling and apoptosis. Mitochondrial dysfunction is implicated in a wide range of diseases and pathological conditions (2, 3, 4). A variety of assays—respirometry—fluorescence probes for assessing $\Delta\Psi_m$ (e.g., JC-1 [5,5',6,6'-tetrachloro-1,1',3,3'-tetraethylbenzimidazo-lylcarbocyanine iodide], TMRM [tetramethylrhodamine methyl ester]), and imaging quantify mitochondrial parameters, but most of them either require exogenous labels or provide population averages.

Consequently, a label-free approach that resolves the functional state of individual mitochondria remains an unmet need. Fluorescent techniques are widely used to assess mitochondrial properties in intact cells and in isolated mitochondria. These techniques provide measurements of $\Delta\Psi_m$, reactive oxygen species production, and network architecture/connectivity (5, 6, 7). Many mechanistic assays are therefore performed on isolated organelles including different $\Delta\Psi_m$- and reactive oxygen species–sensitive fluorescent dyes, electron microscopy, and selective electrodes for oxygen, pH, TPP$^+$ ($\Delta\Psi_m$ measurement), and Ca$^{2+}$ (8, 9, 10, 11). While powerful, probe-based approaches rely on exogenous dyes and careful calibration (e.g., JC-1 quantified as a red/green ratio; TMRM mean pixel intensity) and they can introduce concentration- and phototoxicity-dependent artifacts and typically reflect population averages rather than single-organelle behavior. The activity of specific ion channels and exchangers can be studied using the patch-clamp technique (12, 13, 14). This method (typically mitoplast patch clamp) is technically demanding at submicron scales and at low throughput, and primarily reports electrical properties. A popular method for studying parameters of single cells and occasionally of isolated mitochondria is flow cytometry, including FACS (15). The advantage of FACS is the ability to analyze many objects at high throughput with multiparametric assessment of several characteristics, such as $\Delta\Psi_m$, oxidative stress, and size surrogates (FSC/SSC) (16, 17). However, probe-based cytometry has limitations: potential-sensitive dyes accumulate in the inner membrane or matrix and can affect mitochondrial function. For example, rhodamine derivatives can induce phototoxicity (18), and safranin may inhibit complex I in neuronal cells (19). Accurate readouts further require careful

[1]Department of Immunology and Regenerative Biology, Weizmann Institute of Science, Rehovot, Israel   [2]Department of Chemical Research Support, Weizmann Institute of Science, Rehovot, Israel   [3]Kurchatov Complex of NBICS-Technologies, NRC "Kurchatov Institute," Moscow, Russia

Correspondence: atan.gross@weizmann.ac.il

calibration, whereas suboptimal dosing or illumination can cause signal saturation or toxicity. Low concentrations of dyes are used whenever possible, but the sensitivity to side effects of dyes on mitochondria from different tissues and organisms, in normal and pathological conditions, varies greatly and unpredictably, which can lead to misleading results. The artifacts can also be induced by the movement of dyes between quenched (hydrophobic) and unquenched (hydrophilic) compartments (20). Finally, flow cytometry provides no spatial (suborganelle) resolution; isolated mitochondria register as near-point events close to the detection threshold, which increases measurement variance and emphasizes population distribution over single-organelle behavior. Thus, probing the mitochondria at a higher level of resolution addresses the need to map out biological diversity even within a single cell. This not only avoids network-level coupling but could in principle be used to analyze rare dysfunctional subpopulations.

Atomic force microscopy (AFM) is a powerful method for studying biological samples (21). Typically, a micrometer-scale cantilever with a nanometer-sized, sharp tip is used for biological applications (22). The tip can scan the surface, line by line, and generate topography (z-height) maps. AFM imaging can be performed in very gentle modes (e.g., tapping at or below the resonance frequency) to avoid damage to soft samples, in both air and liquid environments (23, 24, 25). AFM can be used to measure not only sizes of nanoscale objects and surface roughness, but also the elastic properties of the sample via force spectroscopy and contact-mechanics models (26, 27). The height measurement can serve as a proxy for volume/osmotic state. In our implementation, topography is obtained by scanning in tapping mode under feedback control, and fluctuation spectra are recorded by gently holding the probe at a fixed position on a selected mitochondrion under constant-amplitude feedback and sampling the time-resolved height fluctuations derived from the Z-piezo signal (see the Materials and Methods section). By applying AFM for imaging, mechanical analysis, and frequency-resolved height fluctuations, we obtain multimodal, single-organelle analysis in a liquid environment under label-free conditions.

Our proof-of-principle experiments demonstrate that AFM measurements can distinguish different mitochondrial functional states under defined bioenergetic manipulations, with a scalar metric (power of fluctuations integrated from DC to 20 Hz) (28) that covaries with membrane potential ($\Delta\Psi_m$) measured by independent fluorescence methods. Moreover, AFM features delineate genotype-dependent phenotypes. Notably, mitochondria isolated from all three mouse embryonic fibroblast knockouts, *MTCH2−/−*, *MFN1−/−*, and *MFN2−/−*, used in our study, showed a mechanically stiff state as compared to mitochondria isolated from WT MEFs; however, the *MTCH2* knockout mitochondria showed elevated low-frequency fluctuations, whereas the *MFN1* and *MFN2* knockout mitochondria showed decreased low-frequency fluctuations. This single-organelle, label-free nanomechanical framework complements established assays and motivates further validation across additional models and patient-derived material.

# Results

## Mitochondrial height fluctuations measured by AFM correlate with $\Delta\Psi_m$ measured by potentiometric fluorescent dyes

In this study, we used isolated mouse liver mitochondria (MLM), a well-established model system for bioenergetic studies (29). A representative AFM image of MLM obtained in liquid tapping mode is shown in Fig 1A. The purity and integrity of the MLM preparations were confirmed via confocal microscopy using the $\Delta\Psi_m$-sensitive dye TMRM, alongside carbonyl cyanide m-chlorophenyl hydrazone (CCCP), a protonophore used here as a depolarization control (Fig S1A and B). Orthogonal quality controls included transmission electron microscopy (TEM) to verify ultrastructure and Seahorse respirometry to quantify oxygen consumption (Fig S1C, F, and G). $\Delta\Psi_m$ was assessed independently with JC-1 (red/green ratio) and TMRM (ImageStream/IDEAS mean pixel intensity), and AFM fluctuation metrics were correlated with these readouts (see the Materials and Methods section). High-resolution AFM of chemically fixed and dried mitochondria provided surface topography to verify overall morphology and sample integrity (Fig S1D and E). These images give higher resolution, revealing fine, nm-level structure at the surface. As expected, CCCP and ADP, an activator of phosphorylation, increased oxygen consumption rate (OCR) under their respective assay conditions, whereas rotenone and antimycin A, inhibitors of complex I and complex III, respectively, markedly suppressed OCR (Fig S1F and G).

Using AFM topography measurements such as shown in Fig 1A as a guide, the AFM probe was positioned on the surface of individual mitochondria, and the Z-piezo displacement (dynamic height) was recorded for 10 s under constant-amplitude feedback control to monitor nanoscale height fluctuations. The analysis workflow is illustrated in Fig 1B, where the AFM height signal was digitized and analyzed by fast Fourier transform to obtain the power spectral density (PSD). The low-frequency range (DC to 20 Hz) was integrated to assess mitochondrial responses to different functional states while avoiding environmental/cantilever resonances and the high-frequency noise floor.

The power spectra showed characteristic signatures to various respiratory substrates and inhibitors, consistent with their established pharmacology (Fig 1C). Specifically, responses to malate + glutamate (complex I substrates), rotenone (a complex I inhibitor), and CCCP (protonophore/uncoupler) are shown in Fig 1D and E. Further responses to succinate + rotenone (a complex II substrate with complex I blocked to prevent reverse electron flow), ADP (an activator of phosphorylation), and antimycin A (a complex III inhibitor) are presented in Fig 1F and G. Additional analyses of TMPD + ascorbate (complex IV substrates) and azide (an inhibitor of cytochrome-c oxidase, complex IV) are shown in Fig S1H and I, and spectra for the additives alone with no mitochondria are presented in Fig S1J. Under all conditions, energized states exhibited higher low-frequency power (DC to 20 Hz) and characteristic band-limited features, whereas inhibition/uncoupling flattened the spectra and reduced integrated power (see the Materials and Methods section for analysis details).

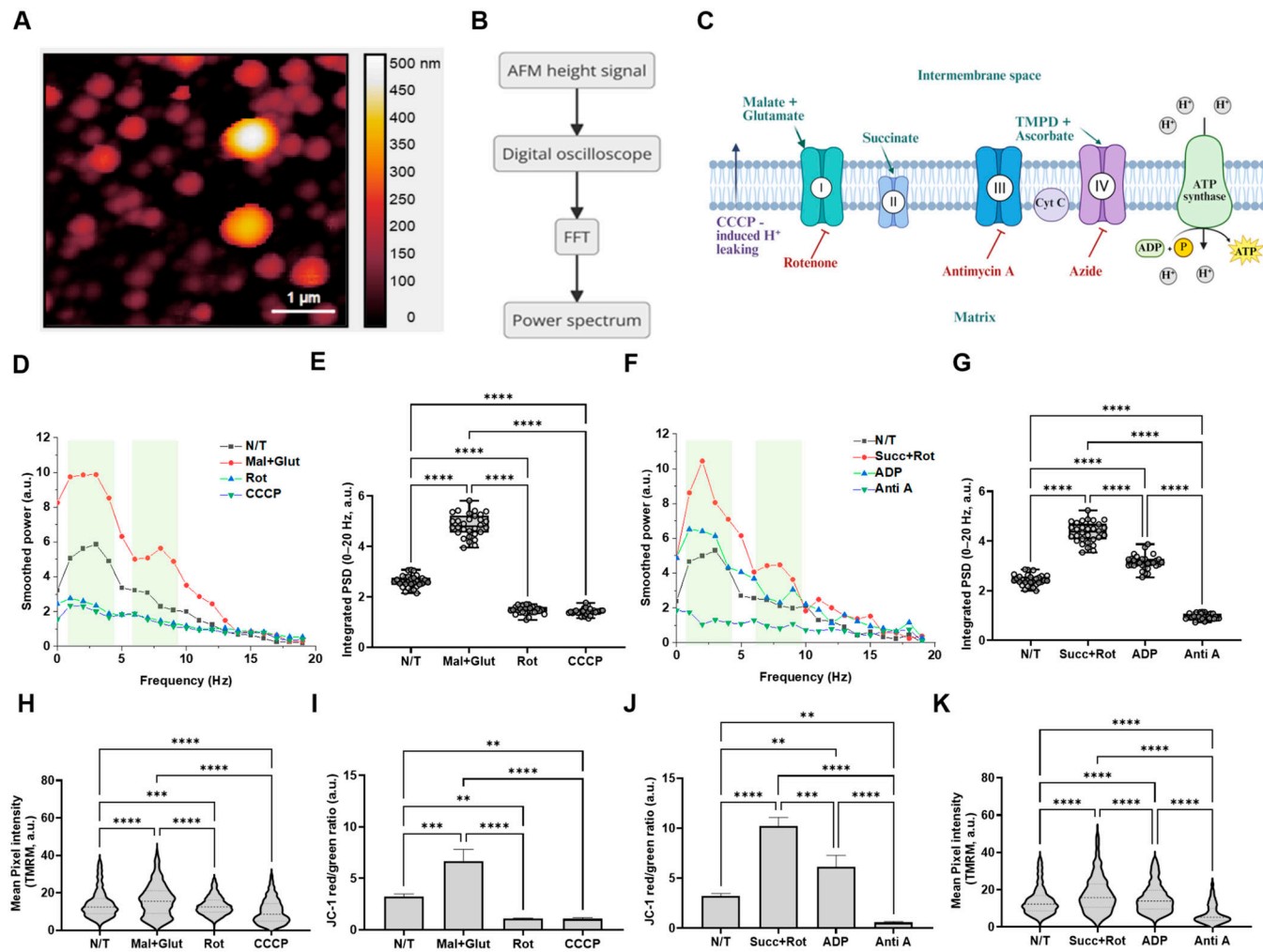

**Figure 1. AFM low-frequency power (P$_{0-20}$) tracks mitochondrial ΔΨ$_m$ across substrates and inhibitors and agrees with dye-based readouts.**
**(A)** Representative AFM topography of isolated mouse liver mitochondria adsorbed on poly-L-lysine–coated mica; liquid tapping mode (scale bar, 1 µm; color denotes height). **(B)** Workflow for transforming AFM height traces into power spectra: digital acquisition, detrending, fast Fourier transform, and spectral analysis/integration. **(C)** Schematic of the electron transport chain with substrates/inhibitors used here (green/red). Succinate + rotenone = complex II with complex I blocked; azide = complex IV inhibitor. **(D)** Mean smoothed power spectral density (PSD; 1-Hz bins) for complex I–linked conditions (N/T, malate + glutamate, rotenone, CCCP). Light-green bands highlight resonances whose intensity varies with energetic state. **(D, E)** Integrated low-frequency power (0–20 Hz, a.u.) for the conditions in (D) (each point represent data for a single mitochondrion). **(F)** Mean smoothed PSD (1-Hz bins) for complex II–linked conditions (N/T, succinate + rotenone, ADP, antimycin A). Light-green bands as in (D). **(G)** Integrated low-frequency power (0–20 Hz, a.u.) for the conditions in (F) (each point = data for a single mitochondrion). **(H)** TMRM mean pixel intensity (a.u.; ImageStream imaging flow cytometry; IDEAS mean pixel = background-subtracted average pixel intensity within the Object mask). Violin plots show event-level distributions; center line = median; dotted lines = quartiles (IQR). **(I, J)** JC-1 red/green ratio (a.u.) by top-read microplate fluorimetry. Signals from red (~540–545/590–595 nm) and green (~485/530 nm) channels were background-subtracted with dye-only blanks; ratios were computed as F590/F530 (see the Materials and Methods section). **(H, K)** TMRM mean pixel intensity (a.u.; ImageStream/IDEAS). For PSD curves (D, F), the SEM at each frequency is smaller than the line width and is omitted; hypothesis testing is performed on the integrated 0- to 20-Hz power shown in (E, G). Panels (E, G): per-mitochondrion scatter; n = 30 mitochondria per condition (three biological replicates). Statistics were performed with one-way ANOVA with Tukey's multiple comparisons; significance is denoted as ns, **$P ≤ 0.01$, ***$P ≤ 0.001$, ****$P ≤ 0.0001$; exact $P$-values are reported in Table S2. For TMRM violins (H, K), event-level distributions are pooled for visualization only; statistics were performed at the biological replicate level (N = 3) with one-way ANOVA/Tukey's test; exact n/tests/$P$ are reported in Table S2. For JC-1 bar plots (I, J), values were averaged within biological replicates before testing; N = 3; one-way ANOVA/Tukey's test; exact $P$ is reported in Table S2. Cross-assay correlations (N/T, Mal + Glut, Rot, CCCP) were performed. JC-1 ratio versus integrated PSD: Pearson's r = 0.909 (exact permutation $P = 0.042$); Spearman's $ρ = 1.00$ ($P = 0.083$); Kendall's $τ = 1.00$ ($P = 0.083$). TMRM mean pixel versus integrated PSD for all conditions: Pearson's r = 0.860 ($P = 0.167$); Spearman's $ρ = 1.00$; Kendall's $τ = 1.00$. Abbreviations: N/T, not treated; Mal + Glut, malate + glutamate; Rot, rotenone; CCCP, carbonyl cyanide m-chlorophenyl hydrazone; Succ + Rot, succinate + rotenone; Anti A, antimycin A. Source Data: per-mitochondrion values underlying all plots are provided in the accompanying Source Data file.
Source data are available for this figure.

To validate the AFM-based assessment of mitochondrial function, we compared the integrated power from AFM spectra with measurements of ΔΨ$_m$. Across conditions, ΔΨ$_m$ obtained by both probes covaried with the integrated power (compare Fig 1E with Fig 1H and I, Fig 1G with Fig 1J and K, and Fig S1I with Fig S1K and L).

Notably, the addition of ADP, which stimulates phosphorylation, resulted in a moderate decline in $\Delta\Psi_m$ (by ~10–20%) mirrored by a decrease in integrated power, consistent across all three methods. Conversely, treatment with CCCP caused an almost complete collapse of $\Delta\Psi_m$ and a near-baseline integrated power.

Although all three methods demonstrated qualitative agreement, the quantitative magnitudes of the changes varied, indicating that calibration would be required for direct quantitative assays. Nonetheless, for functional phenotyping or functional assessment purposes, qualitative or semiquantitative analyses (e.g., percentage of control) are often sufficient. These results show that the AFM-derived integrated low-frequency power correlates with $\Delta\Psi m$ across our perturbations, providing a label-free, indirect readout that complements established fluorescence-based measurements. The details of the frequency response are depicted qualitatively in the power spectra (Fig 1D and F), and the total integrated power up to 20 Hz is summarized with individual-mitochondrion distributions in plots (Fig 1E and G).

## Comparison between elastic modulus and mitochondrial height across different bioenergetic manipulations

A distinctive advantage of AFM over most other methods is its ability to simultaneously analyze surface topography and elastic properties. When imaging fields containing multiple mitochondria (Fig 1A), AFM provides nanometer-scale precision in the vertical (Z) measurement of surface height.

Elastic properties (Young's modulus) are acquired concurrently with topography via force mapping, allowing direct correlation between them. Besides functional readouts, modulus maps also help discriminate mitochondria from similarly sized non-mitochondrial particles (e.g., protein/lipoprotein aggregates or vesicles) that differ in rigidity.

Fig 2 extends the analysis of mitochondrial height and stiffness under the same pharmacological conditions as in Figs 1 and S1. Panels 2A and B show representative Young's modulus maps: panels 2C, D, E, F, G, H report single-mitochondrion distributions for height (C, E, G) and Young's modulus (D, F, H). Energization with malate + glutamate increased height with a concomitant decrease in stiffness relative to untreated mitochondria (Fig 2C and D). Dissipating $\Delta\Psi_m$ with CCCP or inhibiting the respiratory chain with rotenone reduced height and increased stiffness (Fig 2C and D). With TMPD + ascorbate, height exceeded ADP and azide while ADP was comparable to untreated mitochondria, whereas stiffness was lowest for untreated mitochondria and TMPD + ascorbate, increased with ADP, and was highest with azide (Fig 2E and F). When feeding succinate + rotenone, height increased then decreased with addition of both ADP and antimycin A yet ADP was comparable to untreated mitochondria. Addition of succinate + rotenone increased stiffness while addition of ADP reduced it to a level comparable to untreated mitochondria, then addition of antimycin A increased stiffness to the highest level (Fig 2G and H). Per-condition directions of change (↑/↓ relative to untreated mitochondria) are summarized in Table S1.

A decrease in $\Delta\Psi_m$ caused by the protonophore CCCP or each of the respiratory chain inhibitors, rotenone, azide, and antimycin A, led to a reduction in mitochondrial height (Fig 2C, E, and G,

respectively), whereas the elastic modulus/stiffness was increased (Fig 2D, F, and H, respectively). As one might expect from the results described above, the addition of succinate with rotenone resulted in a "hybrid" behavior—an increase in both mitochondrial height and elastic modulus/stiffness (Fig 2G and H, respectively).

## Changes in integrated power are not due to changes in mitochondrial swelling

Apparent covariation between integrated power and height in some conditions raises the question of whether height (matrix volume) itself depends on $\Delta\Psi_m$, or whether only fluctuation amplitude does. Ultrastructural studies have reported associations between mitochondrial energetic state and matrix configuration/volume (30, 31); however, TEM provides static, correlative snapshots and does not directly quantify $\Delta\Psi m$, and swelling responses are condition-dependent (32, 33). Moreover, increases in mitochondrial volume have also been reported upon depolarization/uncoupling (34, 35). Classical ultrastructural work shows that matrix volume is dynamic and responds to ion fluxes and changes in $\Delta\Psi_m$, with swelling occurring under conditions that favor cation/water influx into the matrix (e.g., $K^+$ uptake). The observed changes in size and elastic properties across conditions indicate that mitochondrial physical properties vary with functional state. Because changes in $\Delta\Psi_m$ can be accompanied by ion cycling, we considered whether AFM readouts might reflect swelling-related ion fluxes rather than electron transport modulation per se. To isolate the contribution of osmotic swelling, we used the $K^+$ ionophore valinomycin, which promotes matrix $K^+$ influx with osmotically coupled water entry and can partially dissipate $\Delta\Psi m$ via the $K^+$ cycle (36). To identify conditions under which changes in integrated power can be interpreted as changes in $\Delta\Psi_m$, it was necessary to check how this and other parameters measured by AFM are related to changes in the mitochondrial matrix volume.

The integrated power of mitochondrial height fluctuations decreased significantly upon valinomycin treatment (Fig 3A), consistent with a shift toward a lower $\Delta\Psi_m$ state relative to Mal + Glut. This was accompanied by an increase in mitochondrial height (Fig 3B), consistent with osmotic matrix swelling driven by enhanced $K^+$ influx. This height increase was accompanied by mechanical softening, reflected by a decrease in Young's modulus (Fig 3C). CCCP induced a robust loss of $\Delta\Psi_m$, as indicated by reduced TMRM intensity and a decreased JC-1 red/green ratio (Fig 3D and E). In contrast, valinomycin produced a more modest and assay-dependent change in $\Delta\Psi_m$: TMRM decreased relative to Mal + Glut and relative to NT (one-way ANOVA with Tukey's multiple comparisons; Val versus Mal + Glut, $P \le 0.0001$; Val versus N/T, $P \le 0.0001$), whereas the JC-1 red/green ratio did not significantly differ between N/T and Val (Fig 3D and E). Group means ± SEM and exact multiple comparison statistics are provided in Table S2 and summarized in Table S1. These AFM data show distinct reduction in $\Delta\Psi_m$ but not total collapse. In contrast, CCCP provides the clearest perturbation linking $\Delta\Psi_m$ loss to the AFM readouts, because it produced the strongest decrease in $\Delta\Psi_m$ across both fluorescent assays (Fig 3D and E).

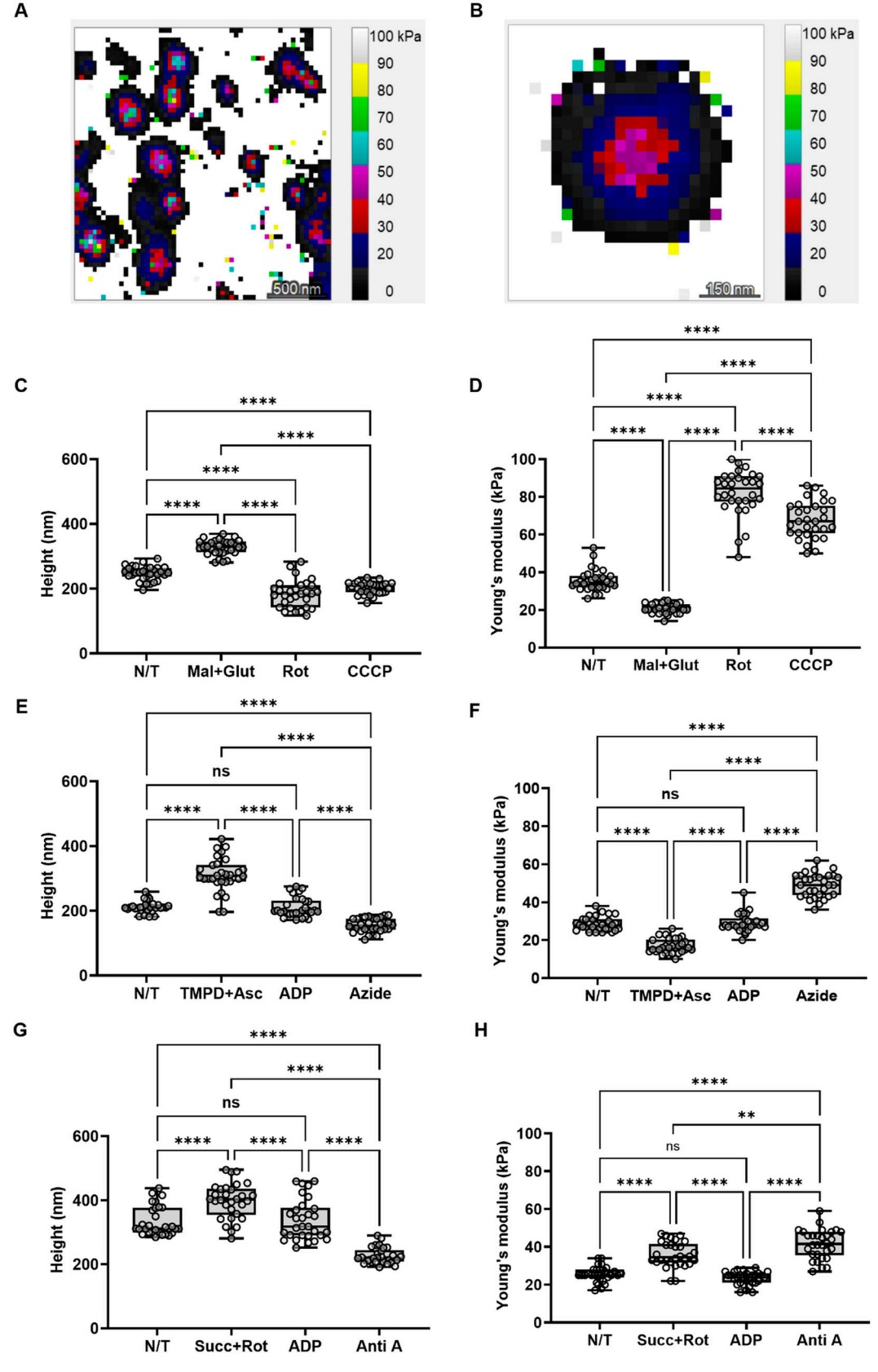

Pearson's correlation coefficients for the valinomycin data of Fig 3 reveal that neither mitochondrial height nor Young's modulus strongly correlates with changes in integrated power across the conditions tested (all |r| < 0.3).

This indicates that swelling/softening does not covary with the $\Delta\Psi_m$-linked changes in fluctuation power. The opposite direction of changes (height increases, whereas integrated power decreases under valinomycin) further argues that integrated power does not mirror swelling-driven height changes. The inverse response relative to Figs 1 and 2—height increases, whereas integrated power decreases under valinomycin, demonstrates that integrated power is not a proxy for swelling. Thus, the correlation of integrated power with $\Delta\Psi_m$, but not with swelling, is confirmed.

### AFM analysis detects functional defects in mitochondria

To assess whether AFM features resolve functional defects in mitochondria, we examined mouse liver mitochondria (MLM) isolated from *MTCH2* liver-conditional knockout mice. MTCH2 is an outer mitochondrial membrane (OMM) protein implicated in mitochondrial metabolism, dynamics, and apoptosis (37, 38, 39, 40); recent work identifies it as a protein insertase, shaping the OMM proteome (41), and/or a phospholipid scramblase, shaping the OMM phospholipid composition (42). We asked whether single-organelle AFM measurements capture the mechanical–functional consequences of MTCH2 loss.

AFM revealed changes in the mechanical and bioenergetic properties of *MTCH2* knockout (MKO) mitochondria compared with WT mitochondria. Specifically, MKO mitochondria exhibited increased integrated power (Fig 4A), in accordance with an elevation in $\Delta\Psi_m$ (Fig 4B and C) and the published literature (40). In addition, MKO mitochondria displayed increased Young's modulus (Fig 4D), along with a decrease in mitochondrial height (Fig 4E).

Notably, the combination of higher integrated fluctuation power together with higher $\Delta\Psi_m$, increased the stiffness, and reduced height in *MTCH2*-deficient mitochondria differs from the inhibitor/uncoupler profiles (Figs 1 and 2), which lower both integrated power and $\Delta\Psi_m$. Therefore, MTCH2 deletion does not phenocopy uncoupling or electron transport chain inhibition; rather, it is consistent with a hyperpolarized, mechanically tightened, and osmotically condensed state (see the Discussion section). This interpretation is concordant with prior reports of altered bioenergetics in *MTCH2*-deficient systems and with our mechanistic rationale that changes in the OMM proteome and/or OMM phospholipid composition can secondarily modulate $\Delta\Psi_m$-coupled nanomotions and surface nanomechanics.

### AFM nanomechanics reveals unique characteristics for mitochondria isolated from *MTCH2−/−*, *MFN1−/−*, or *MFN2−/−* mouse embryonic fibroblasts

To assess function-defect separability and generality beyond liver, we analyzed mitochondria deficient in one of three different genes, which were isolated from immortalized mouse embryonic fibroblasts (MEFs).

Mitochondria isolated from *MTCH2*-deficient MEFs (*MTCH2* KO), as compared to mitochondria isolated from WT MEFs, exhibited the same unique characteristics as MKO mitochondria isolated from liver (Fig 5A–D; compare with Fig 4A–E). We also analyzed mitochondria isolated from MEFs deficient in one of the two mitochondrial pro-fusion GTPases, MFN1 and MFN2 (*MFN1* KO and *MFN2* KO). MFN1 and MFN2 are localized to the OMM and are the regulators of OMM fusion (43). Intriguingly, like the *MTCH2* KO mitochondria, both *MFN1* KO and *MFN2* KO mitochondria showed a mechanically stiff state (Fig 5E). However, unlike the *MTCH2* KO mitochondria, both *MFN1* KO and *MFN2* KO mitochondria showed decreased integrated power (Fig 5G), consistent with the lower JC-1 red/green ratios (Fig 5H) and published literature (44). *MTCH2* KO mitochondria were markedly smaller than WT (height; Fig 5B), and *MFN1* KO mitochondria were also smaller than WT (height; Fig 5F), whereas *MFN2* KO mitochondria did not differ in height from WT (Fig 5F).

## Discussion

By tracking nanoscale height fluctuations ("noise") of an individual organelle over time, AFM provides a label-free readout that covaries with bioenergetic manipulations. AFM was first used to monitor activity in a biological system by looking at height fluctuations on a protein confined to a surface upon exposure to its enzymatic substrate (45). Mitochondria represent a more complex system than single proteins. Accordingly, the AFM method employed here measured several characteristics including topography (height) and elastic modulus, as well as the frequency response of height fluctuations.

All AFM measurements in this study are performed on isolated mitochondria adhered to solid support under ex vivo conditions.

**Figure 2. AFM height and Young's modulus of isolated mitochondria across respiratory states.**
**(A)** Representative Young's modulus map of multiple isolated mouse liver mitochondria adhered to a poly-L-lysine–coated mica surface (color bar, kPa). **(B)** High-resolution Young's modulus map of a single mitochondrion showing spatial stiffness variations (color bar, kPa). **(C, D)** Complex I–linked conditions (N/T, malate + glutamate, rotenone, CCCP): (C) height (nm) and (D) Young's modulus (kPa) for individual mitochondria. **(E, F)** Complex IV–linked conditions (N/T, TMPD + ascorbate, ADP, azide): (E) height and (F) Young's modulus. **(G, H)** Complex II–linked conditions (N/T, succinate + rotenone, ADP, antimycin A): (G) height and (H) Young's modulus. AFM acquisition and analysis were performed. Height and force mapping were performed in liquid; Young's modulus was obtained from force–distance curves by Hertz-type fits (Poisson's ratio 0.5) over the low-indentation regime (see the Materials and Methods section). Each dot represents data for a single mitochondrion; n = 30 mitochondria per condition drawn from N = 3 biological replicates. Bars show the mean ± SEM. Statistics were performed with one-way ANOVA with Tukey's multiple comparisons test; significance: ns, **$P \leq 0.01$, ****$P \leq 0.0001$. Exact n/tests/p are provided in Table S2. Abbreviations: N/T, not treated; Mal + Glut, malate + glutamate; Rot, rotenone; CCCP, carbonyl cyanide m-chlorophenyl hydrazone; TMPD + Asc, N,N,N′,N′-tetramethyl-p-phenylenediamine + ascorbate; Succ + Rot, succinate + rotenone; Anti A, antimycin A. Panels (C, D, E, F, G, H) represent three independent condition sets acquired in separate experimental sessions/preparations. Each set includes its own session-matched N/T control (C/D: N/T, Mal + Glu, Rot, CCCP; E/F: N/T, TMPD + Asc, ADP, azide; G/H: N/T, Succ + Rot, ADP, antimycin A); therefore, N/T values across panels should not be compared directly. Statistical comparisons are performed within each set against its matched N/T.

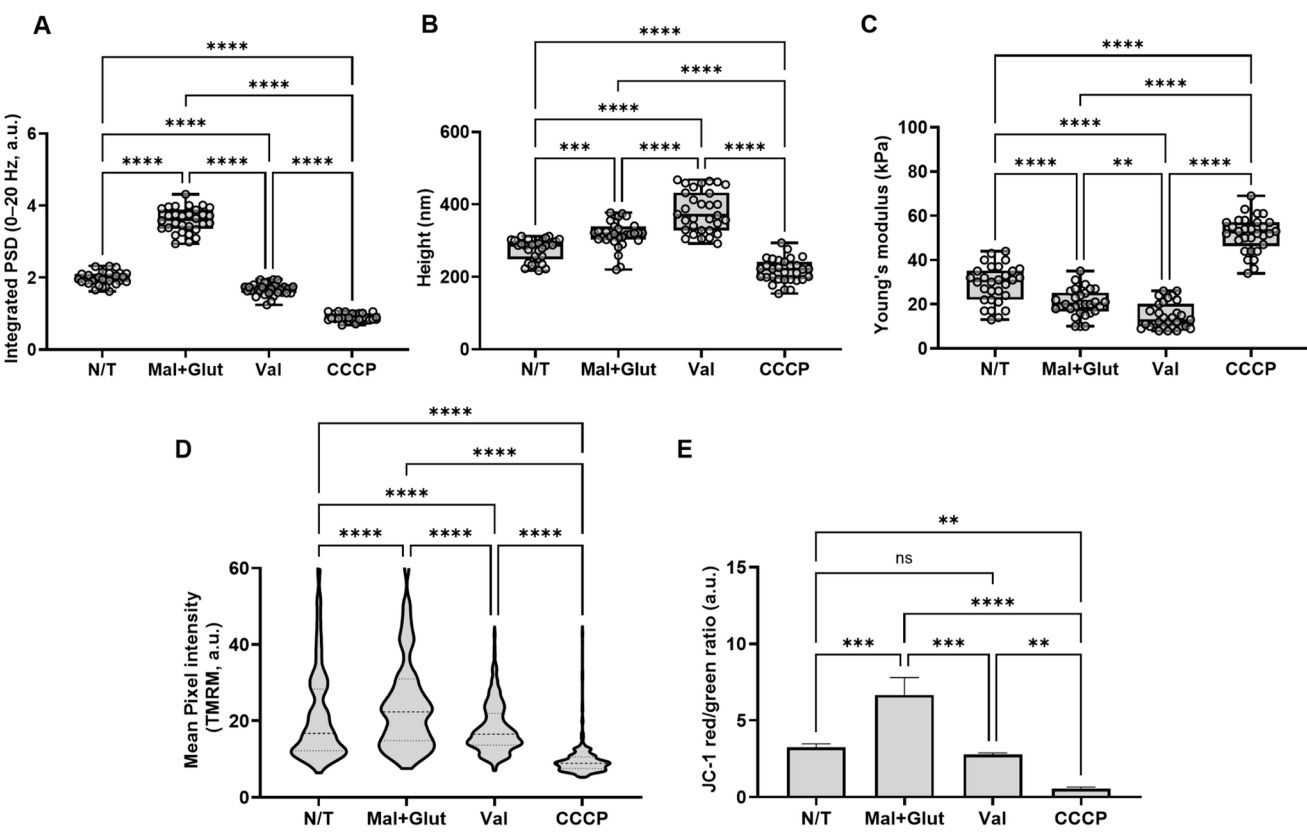

**Figure 3. Valinomycin decouples ΔΨ_m from swelling/softening: AFM $P_{0-20}$ tracks potential, whereas height and stiffness report nanomechanics.**
**(A)** Integrated low-frequency power of AFM height fluctuations (0–20 Hz, a.u.) across conditions (N/T, malate + glutamate, valinomycin, CCCP). $P_{0-20}$ rises with substrate and is reduced by valinomycin, with the lowest values under CCCP. **(B)** Height (nm) of individual mitochondria showing matrix swelling with valinomycin (K⁺ influx/osmotic imbalance). **(C)** Young's modulus (kPa) indicating mechanical softening with valinomycin and CCCP. **(D)** TMRM mean pixel intensity (a.u.; ImageStream imaging flow cytometry, IDEAS mean pixel = background-subtracted average pixel intensity within the Object mask). Violin plots show event-level distributions; center = median; dotted lines = quartiles (IQR). **(E)** JC-1 red/green ratio (a.u.) by top-read microplate fluorimetry. Red (~540–545/590–595 nm) and green (~485/530 nm) channels were background-subtracted with dye-only blanks; ratios were computed as F590/F530 (see the Materials and Methods section). AFM panels (A, B, C): each dot represents data for a single mitochondrion; n = 30 per condition from N = 3 biological replicates; bars show the mean ± SEM. Statistics were performed with one-way ANOVA with Tukey's multiple comparisons test; significance: ns, **P ≤ 0.01, ***P ≤ 0.001, ****P ≤ 0.0001; exact n/tests/P are reported in Table S2. TMRM violin (D): event-level distributions are pooled for visualization only; statistics were performed at the biological replicate level (N = 3); one-way ANOVA/Tukey's test; exact n/tests/P are reported in Table S2. JC-1 (E): values were averaged within biological replicates before statistical testing; N = 3; one-way ANOVA/Tukey's test; exact P is reported in Table S2. Abbreviations: N/T, not treated; Mal + Glut, malate + glutamate; Val, valinomycin; CCCP, carbonyl cyanide m-chlorophenyl hydrazone.

Isolation and handling can perturb mitochondrial physiology and enrich for fragmented, rounded organelles, and removal from the cellular environment eliminates interactions with the cytoskeleton and with other organelles (including ER contact sites) that shape mitochondrial behavior in situ (46). Accordingly, the AFM-derived parameters reported here should be interpreted as comparative ex vivo single-organelle phenotypes rather than a direct representation of mitochondrial physiology within intact cells. "Height" in our assay reflects organelle geometry after isolation and adhesion and is not intended as a proxy for filamentous/network morphology in living cells.

To understand how this information can be related to mitochondrial function, we dissect the components of the noise signal. Power spectra, such as those shown in Fig 1D and F, show an overall trend of decreasing power with frequency, with enhancement at two different frequency ranges (2–4 Hz and 6–9 Hz) superimposed on this background. Both the broad envelope and the prominent peaks are of interest. The background profile can be

described by frequency dependence of $1/f^{\alpha}$ for mitochondria. Inhibited states exhibit flatter, white noise-like spectra ($\alpha < 0.5$), whereas energized states display steeper, pink-like spectra ($\alpha \approx 1$ or higher) (47).

Noise plays an important part in harnessing various physiological regulatory processes occurring at different frequencies. Frequency behavior where the $1/f^{\alpha}$ noise has an exponent $\alpha$ close to 1 is termed "pink" noise and is characteristic of viable systems (48, 49, 50). This behavior is found, for instance, in the human heartbeat. Previous studies have shown that most viable systems have an exponent close to 1. An exponent of zero, otherwise known as "white noise," would result in a "flat" power spectrum with equal energy at all frequencies and is indicative of an aging or unhealthy organism. This small $\alpha$ behavior, indicated by a weak slope in the log–log plot of power versus frequency, is seen after inhibitor additions such as rotenone or antimycin A. This is also the case for control on the mica substrate. These are shown in Fig S2A and B. Interestingly, untreated mitochondria give values very similar to

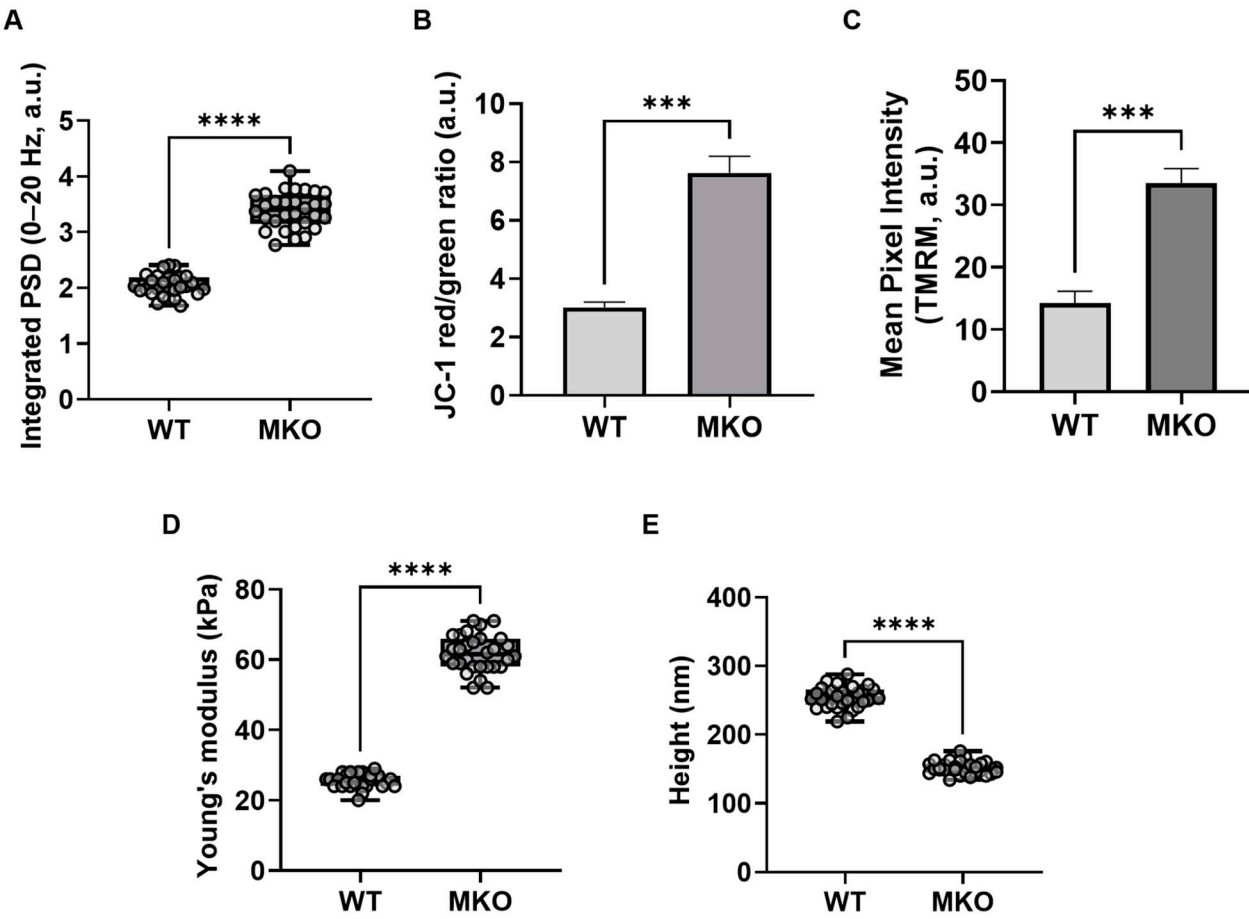

**Figure 4. Biophysical and ΔΨ$_m$-linked differences between mitochondria isolated from WT and *MTCH2* liver-conditional knockout (MKO) mice.**
**(A)** Integrated low-frequency power of AFM height fluctuations (0–20 Hz, a.u.) per mitochondrion (proxy that tracks ΔΨ$_m$). **(B)** JC-1 red/green ratio (a.u.) by top-read microplate fluorimetry. Red (~540–545/590–595 nm) and green (~485/530 nm) channels were background-subtracted with dye-only blanks; ratios were computed as F590/F530 (see the Materials and Methods section). **(C)** TMRM mean pixel intensity (a.u.) by ImageStream imaging flow cytometry (IDEAS mean pixel = background-subtracted average pixel intensity within the Object mask). **(D)** Young's modulus (kPa) from AFM force–distance curves (Hertz-type fits; Poisson's ratio 0.5). **(E)** Height (nm) from AFM topography. **(A, B, C, D, E)** MKO mitochondria exhibit higher ΔΨ$_m$-linked readouts (A, B, C), greater stiffness (D), and smaller height (E) than WT. Statistics and n are reported. AFM panels (A, D, E): each dot represents data for a single mitochondrion; n = 30 per group (10 per biological replicate × N = 3). Bars show the mean ± SEM. Group comparisons by unpaired two-tailed *t* test; significance: ns, ***$P \leq 0.001$, ****$P \leq 0.0001$; exact n/tests/$P$ are reported in Table S2. JC-1 (B) and TMRM (C): values were averaged within biological replicates before testing; N = 3; bars show the mean ± SEM; unpaired two-tailed *t* test; exact $P$ is reported in Table S2. Abbreviations: WT, wild type; MKO, *MTCH2* liver-conditional knockout; N/T, not treated; JC-1, 5,5',6,6'-tetrachloro-1,1',3,3'-tetraethylbenzimidazolylcarbocyanine iodide; TMRM, tetramethylrhodamine methyl ester. WT data shown in this figure are experiment-matched controls acquired in parallel with the indicated condition(s) during the same measurement session; therefore, WT values should not be compared across different figures/experiments. Statistical comparisons are performed within each figure against its matched WT control.

those treated with succinate + rotenone for which complex I is inhibited and complex II activated, around α = 0.9. Activation by malate/glutamate increases α above 1 (Fig S2C and D). Numerical values of α (mean ± SD) for all conditions are reported in Table S3. The second component of these spectra is the distinct enhancement seen at certain frequency ranges. The enhancement over the range 2–4 Hz is seen both for untreated mitochondria and for all substrates that enhance the activity. It is never seen under inhibition. This frequency has been observed previously in ensemble systems. One of these works employed dynamic phase microscopy and associated it with action of the ATP synthase activity (51). A second work, which monitored fluctuations of an AFM cantilever bearing adsorbed mitochondria, found a distinct peak at 1–3 Hz (52). As in our work, these oscillations were suppressed when inhibited by rotenone. Together with prior ensemble measurements (28, 53), our single-organelle data support the interpretation that this low-frequency enhancement is a feature of energized mitochondria in ex vivo preparations. The distinguishing aspect of this work is that we measure single organelles and can correlate the size, modulus, and fluctuation spectrum with the mitochondrial state. The second peak we observed at 6–8 Hz appears under activation and to our knowledge has not been reported at single-organelle resolution; its origin remains to be clarified.

To summarize the fluctuation signal, we integrate the PSD over the DC-20 Hz band to obtain a scalar "integrated power" (RMS amplitude) (54, 55). For all conditions, this metric covaries with ΔΨ$_m$

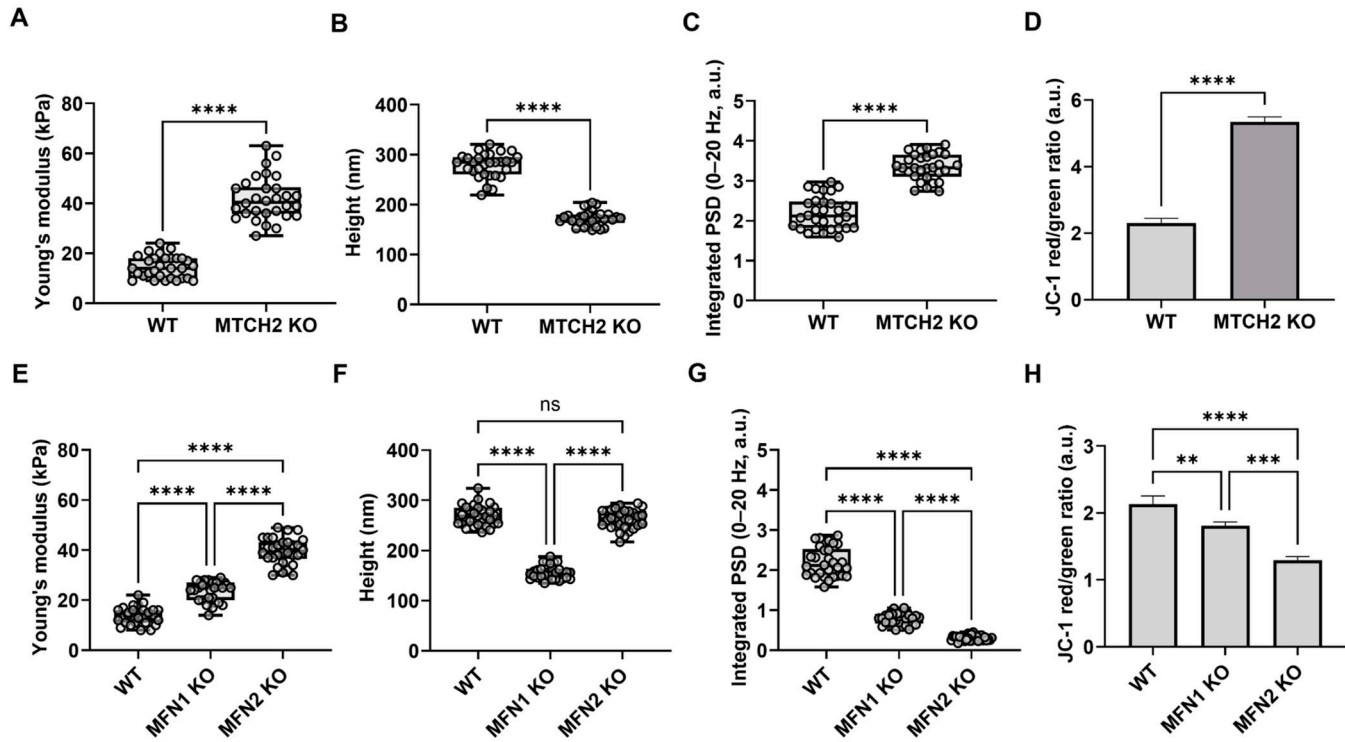

**Figure 5. AFM nanomechanics distinguishes mitochondria isolated from *MTCH2*- and mitofusin-deficient MEFs.**
**(A, B, C, D)** WT versus *MTCH2* KO mitochondria isolated from MEFs: (A) Young's modulus (kPa), (B) height (nm), (C) integrated low-frequency power of AFM height fluctuations ($P_{0-20}$ = 0–20 Hz, a.u.), and (D) JC-1 red/green ratio (a.u.). **(E, F, G, H)** WT versus *MFN1* KO versus *MFN2* KO MEF-derived mitochondria: (E) Young's modulus (kPa), (F) height (nm), (G) integrated low-frequency power ($P_{0-20}$, a.u.), and (H) JC-1 red/green ratio (a.u.). JC-1 assay was performed by top-read microplate fluorimetry; green ~485/530 nm, red ~540–545/590–595 nm channels were background-subtracted with dye-only blanks; ratios were computed as F590/F530. Values were averaged within biological replicates before hypothesis testing (N = 3). Statistics and n for AFM panels (A, B, C, E, F, G): each dot represents data for a single mitochondrion; n = 30 per genotype (10 per biological replicate × N = 3); bars show the mean ± SEM. **(A, B, C, D)** (WT versus MKO): unpaired two-tailed *t* tests. **(E, F, G, H)** (WT versus MFN1 KO versus MFN2 KO): one-way ANOVA with Tukey's multiple comparisons. Significance: ns, **$P \le 0.01$, ***$P \le 0.001$, ****$P \le 0.0001$; exact n/tests/*P* are provided in Table S2. Abbreviations: WT, wild type; *MTCH2* KO, *MTCH2* knockout; *MFN1* KO/*MFN2* KO, *MFN1/2* knockout. WT data shown in this figure are experiment-matched controls acquired in parallel with the indicated condition(s) during the same measurement session; therefore, WT values should not be compared across different figures/experiments. Statistical comparisons are performed within each figure against its matched WT control.

quantified independently by TMRM mean pixel intensity and by the JC-1 red/green ratio (Figs 1 and S1); statistical associations are reported in Methods/Source Data. Although the scalar does not capture the finer spectral features, it provides a simple, robust single-organelle readout that tracks bioenergetic state and complements fluorescence-based assays.

Importantly, valinomycin experiments dissociate matrix volume changes from polarization (56). K⁺ ionophore treatment decreased integrated power and shifted ΔΨm readouts toward lower polarization relative to malate/glutamate (Fig 3D and E), while increasing height and reducing elastic modulus, consistent with osmotic swelling and mechanical softening (Fig 3). Because valinomycin increases height while lowering integrated power, it demonstrates that integrated power is not a proxy for swelling; instead, it more closely tracks polarization. Applying the assay to MKO liver mitochondria revealed a compact, mechanically tightened, hyperpolarized state: integrated power and ΔΨm were higher, stiffness increased, and height decreased relative to WT, a combination that differs from inhibitor/uncoupler profiles which lower both integrated power and ΔΨm. These data indicate that MTCH2 deletion does not phenocopy uncoupling or electron

transport inhibition but yields a distinct mechanobioenergetic phenotype.

To test generalizability across preparations and defect specificity, we extended the analysis to mitochondria isolated from immortalized MEFs (43). In MEF-derived *MTCH2* KO mitochondria, the same pattern observed in liver was reproduced: integrated fluctuation power and ΔΨm were higher, stiffness increased, and height decreased relative to WT (Fig 5A–D; unpaired two-tailed *t* tests). In contrast, mitofusin deficiency produced an opposite coupling between nanomechanics and fluctuations. Across WT, *MFN1* KO, and *MFN2* KO, stiffness increased stepwise, whereas integrated power and JC-1 ratios declined in the same order; geometrically, *MFN1* KO mitochondria were markedly smaller than both WT and *MFN2* KO, whereas WT and *MFN2* KO did not differ in height (Fig 5E–H). In our data, this MFN2-related phenotype is reflected by reduced integrated power (Fig 5G) and lower JC-1 red/green ratios (Fig 5H), together with increased elastic modulus (Fig 5E) and no detectable change in height relative to WT (Fig 5F). This pattern is consistent with prior reports linking MFN2 loss to altered mitochondrial bioenergetics and, in some systems, reduced mitochondrial coenzyme Q levels

with partial rescue by coenzyme Q10 supplementation ([57], [58], [59]); however, we did not directly assay ER–mitochondria contacts or the coenzyme Q pool in the present study, so these mechanisms are discussed only as possible contributors ([60], [61]). It demonstrates that AFM-based measurements can differentiate between molecularly distinct defects rather than reporting a generic stiffness–activity trade-off. Again, as in the *MTCH2* KO mitochondria, both *MFN1* KO and *MFN2* KO mitochondria also show increased stiffness, likely pointing to mitochondrial malfunction.

Collectively, three AFM-derived features, Young's modulus, height, and integrated fluctuation power, robustly separate genotypes at single-mitochondrion resolution. Importantly, the coupling between stiffness and fluctuation power is genotype/function-dependent. AFM thus resolves mechanistically distinct defects and complements fluorescence-based bioenergetic readouts. The AFM approach is orthogonal to fluorescence-based methods and can be integrated with them, in addition to adding the physical aspect (stiffness), to improve single-organelle functional phenotyping. Further work across additional models and patient-derived material, alongside increased throughput and refined biophysical modeling of fluctuation origins, will be important to define scope and utility. Our results underscore the value of integrating orthogonal assays for a comprehensive understanding of the biophysical and biochemical processes occurring in mitochondria. Specifically, label-free AFM nanomechanics at the single-organelle level quantifies topography (height), Young's modulus, and integrated fluctuation power (DC to 20 Hz); this scalar fluctuation metric covaries with $\Delta\Psi_m$ measured independently and remains distinct from matrix swelling (valinomycin), thereby separating polarization from volume effects. The approach provided logical results for both liver- and cell-derived mitochondria and resolves genotype/defect-dependent phenotypes (e.g., MTCH2 loss versus MFN1/MFN2 loss in MEFs), complementing fluorescence-based readouts and respirometry.

# Materials and Methods

### Animals and ethical compliance

All animal work was conducted in accordance with European guidelines for the care and use of laboratory animals (Directive 2010/63/EU) and under protocols approved by the institutional animal ethics committee of the Weizmann Institute of Science. Animals were housed and handled according to standard welfare procedures; reporting follows ARRIVE recommendations where applicable.

### Mitochondrial isolation from mouse liver

Mitochondria were isolated from freshly excised mouse liver by standard differential centrifugation. The tissue was rinsed in ice-cold HIM buffer (pH 7.5; see the Reagents and solutions section) supplemented with 0.2% fatty acid–free BSA (A7030; Sigma-Aldrich). Unless stated otherwise, all steps were performed on ice or at 4°C.

Liver was minced in ~3 ml HIM + 0.2% BSA and homogenized with 6–8 strokes in a 7-ml glass homogenizer (885300-0007; Kimble). The homogenate was diluted with ice-cold HIM and centrifuged at 600$g$, 10 min, 4°C (fixed-angle rotor) to remove nuclei and debris. The supernatant was transferred to a fresh tube and centrifuged at 7,000$g$, 15 min, 4°C to pellet mitochondria. The pellet was gently resuspended in BSA-free HIM and clarified at 700$g$, 10 min, 4°C (discard pellet), and the supernatant was centrifuged again at 7,000$g$, 15 min, 4°C. The final mitochondrial pellet was resuspended in 200 $\mu$l ice-cold, BSA-free HIM for immediate use. Protein concentration was determined where indicated (e.g., BCA), and preparations were kept on ice throughout.

### MEF cell lines and culture

Immortalized MEFs used in this study were WT (control), *MTCH2* KO, *MFN1* KO, and *MFN2* KO. All MEF lines were generated and maintained in the laboratory of Atan Gross (Weizmann Institute of Science). *MTCH2* KO MEFs were generated by Cre recombinase–mediated deletion in MTCH2 floxed (MTCH2^F/F) MEFs, as previously described ([37]). *MFN1* KO and *MFN2* KO MEFs were derived from E10.5 embryos of $Mfn1^{-/-}$ or $Mfn2^{-/-}$ mice and immortalized by retroviral SV40 large T antigen expression as described previously ([43]). All lines were handled in parallel under identical conditions.

Cells were maintained at 37°C in a humidified 5% $CO_2$ incubator in DMEM, high glucose (4.5 g/liter; without sodium pyruvate), supplemented with 10% FBS, 2 mM L-glutamine (or GlutaMAX), 2 mM sodium pyruvate, and 1% penicillin–streptomycin (100 U/ml penicillin, 100 $\mu$g/ml streptomycin). Medium was replaced every 2–3 d, and cells were passaged at 70–90% confluence using 0.05% trypsin–EDTA. For mitochondrial isolation, cells were gently collected with a cell scraper in ice-cold PBS (pH 7.4; no protease inhibitors); all subsequent steps were performed on ice.

### Isolation of mitochondria from MEFs

All steps were performed on ice (4°C) using ice-cold HIM buffer (pH 7.5; see the Reagents and solutions section). Monolayers were rinsed with 5–10 ml PBS; 1 ml PBS was added, cells were scraped on ice, and the suspension was transferred to a 1.5- to 2-ml tube. The dish was rinsed with an additional 1 ml PBS and pooled with the first suspension (~2 ml total). Cells were pelleted at 450$g$, 3 min, 4°C; the supernatant was discarded. The pellet was gently resuspended in 200 $\mu$l HIM and mechanically disrupted by passing the suspension through a 25 G needle 15× on ice. Nuclei/debris were removed by centrifuging the mixture at 1,000$g$, 10 min, 4°C; the supernatant (mitochondria) was transferred to a fresh tube. Optionally, the pellet was re-extracted in 200 $\mu$l HIM and the process was repeated at 1,000$g$, 10 min; supernatants were pooled. Crude mitochondria were pelleted at 10,000$g$, 15 min, 4°C. The washed mitochondrial pellet was resuspended in 200 $\mu$l HIM and centrifuged at 1,000$g$, 10 min, 4°C; the supernatant was transferred

to a fresh tube, and the mitochondria were repelleted at 10,000*g*, 15 min, 4°C. The final pellet was resuspended in 100–200 $\mu$l HIM (no BSA), kept on ice, and used immediately. The protein was measured where indicated (e.g., BCA).

### AFM sample preparation (PLL-mica)

Freshly cleaved muscovite mica was coated with 0.01% (wt/vol) poly-L-lysine (PLL; P8920; Sigma-Aldrich) for ≥30 min (up to overnight) at RT, rinsed 3× with ddH$_2$O and once with ice-cold HIM, and used immediately. Isolated mitochondria in BSA-free HIM were deposited onto PLL-mica (enough volume to cover the surface), incubated for 10 min on ice to promote adhesion, and gently washed 3–4× with 200 $\mu$l ice-cold HIM to remove nonadherent material (slow pipetting along the edge to minimize shear). Samples remained submerged in HIM for all live measurements.

### Fixed/dried reference imaging (identity only)

For morphology identity controls (crista visualization), WT mouse liver mitochondria (MLM; N/T/no-additions control; same preparations as used for live AFM measurements) were adhered to PLL-mica, fixed with 0.2% glutaraldehyde for 15 min, rinsed, air-dried, and imaged in air (Fig S1D and E). Fixed/dried preparations were not used for fluctuation analyses.

### AFM

#### Instruments and probes
Two AFM platforms were used: (i) Bruker MultiMode (liquid tapping) for topography and fluctuation time series; and (ii) JPK NanoWizard III (QI mode) for force mapping and Young's modulus. Silicon nitride probes were Nanosensors qp-BioAC-CI (CB2), nominal spring constant ~0.1 N/m, resonance in buffer ~15 kHz, and tip radius ~30 nm. Cantilevers were equilibrated in HIM before use; deflection sensitivity and spring constant were calibrated before each session of force mapping using the contactless calibration built into the JPK software.

#### Fluctuation time series and PSD
Liquid tapping topography was first acquired to identify well-adhered, isolated mitochondria. The tip was then positioned at the organelle apex and lateral scanning stopped, with feedback maintained while recording the height signal. The tapping frequency, ~15 kHz, was three orders of magnitude above the frequencies of interest here, so no coupling is expected between the modes. The Z-piezo displacement (height) signal (units of volts; approximate conversion ≈ 0.2 nm/V) was routed to a PicoScope 3205F and digitized at 10 kS/s for 10 s per mitochondrion/condition. Within each run, the constant-amplitude tapping set-point and feedback gains were held identical; gains were minimized to limit feedback noise while maintaining stable contact and sufficient bandwidth (time constant of 2 ms = 500 Hz) to follow the fluctuation at the frequencies of interest. The response time was verified by scanning rapidly over a 25-nm step under the experimental working conditions and recording the change in height with time on an external scope. Importantly, our analysis and

integration were restricted to a few tens of Hz, well below the feedback-response frequency. Fluctuations were on the order of a few Ångtroms. Signals were determined before spectral analysis. PSD was computed by FFT, binned at 1 Hz, and normalized to the mean amplitude in a featureless 13–20 Hz band, yielding arbitrary units (a.u.) in the plots. The integrated power over 0–20 Hz (trapezoidal rule) was used as the primary summary metric. Per-frequency SEM was negligible relative to line width and is omitted for clarity; uncertainty and statistics are reported for the integrated metric (replicates: n = 30 mitochondria per condition; N = 3 biological replicates unless noted otherwise).

#### *Elasticity mapping (QI mode)*
Quantitative force mapping (5 × 5 $\mu$m$^2$; 128 × 128 pixels) was performed with a maximum normal force ≤ 150 pN and approach/retract speeds ~30/50 $\mu$m·s$^{-1}$. Under these conditions, the penetration depth was always less than 20% of the particle height to avoid substrate influence. At each pixel, force–distance curves were fit to a Hertzian contact model (Poisson's ratio = 0.5; spherical tip radius ~30 nm) in JPK software to obtain Young's modulus. Here, height refers to the topographic height at each pixel (relative to the mica substrate). Particles < ~150 nm in height were excluded. For per-mitochondrion values, pixels from a central ROI were averaged; curves with unstable baselines or poor fits were rejected. After additions, lateral drift typically precluded tracking the same organelle across conditions; therefore, comparisons were made between matched populations measured under identical parameters on the same day. A representative raw force–distance curve and the Hertz fit are shown in Fig S3.

#### *Imaging flow cytometry (ImageStream; TMRM $\Delta\Psi_m$)*
Isolated mitochondria were always maintained on ice in ice-cold HIM buffer (pH 7.5; see the Reagents and solutions section); no fixation was used. The mitochondria were stained with TMRM 0.5 $\mu$M for 10–15 min on ice in the dark (HIM). The data were acquired by immediate analysis on an Amnis ImageStream with identical laser/detector settings across all conditions within a run. Imaging flow cytometry provides a readout as mean pixel intensity (TMRM, a.u.) in IDEAS, defined as the average pixel intensity within the Object mask after background subtraction. Focused singlets were gated by gradient RMS (in-focus) and area/aspect ratio (singlets); debris/aggregates were excluded using bright-field–based masks. Energized (substrates) and depolarized (CCCP) controls were included in each run to verify dynamic range. Statistics were reported for N = 3 biological replicates. Event-level distributions are shown for transparency (≈1,335 events/condition pooled; ~445 per replicate), but inference was performed at the biological replicate level. Where indicated, values were normalized to the within-run vehicle control (=1). Exact tests and *P*-values are reported in the figure legends, with a consolidated summary in Table S2. "TMRM mean pixel intensity (a.u.)" is used as the axis label in figures. Absolute values for mean pixel appear numerically small (≈0–80 a.u.) because of conservative, nonsaturating detector settings and stringent background subtraction for submicron objects; the readout is linear and preserves between-condition separability.

## Microplate fluorimetry (JC-1 ΔΨ$_m$, ratiometric)

All steps were performed on ice in ice-cold HIM buffer (pH 7.5; see the Reagents and solutions section) and protected from light. The mitochondria were loaded with JC-1 5 µM for 10–15 min on ice (microtubes), washed once by centrifuging at ~ 5,500$g$, 10 min, 4°C, and resuspended in ice-cold HIM. Black, flat-bottom 96-well plates with 100 µl per well volume were loaded with equal mitochondrial protein per well. Signals from green (monomer) Ex ~485-nm/Em ~530-nm and red (J-aggregate) Ex ~540- to 545-nm/Em ~590- to 595-nm channels were acquired by top-read microplate fluorimetry and background-subtracted with dye-only blanks (HIM + JC-1; no mitochondria). Ratios were computed as red/green ratio per well (F590/F530; a.u.); where indicated, they were normalized to the within-run vehicle control (=1). Technical replicates were averaged to a single value per biological replicate before testing. Statistics were reported for N = 3 biological replicates, typically, six technical wells per condition. Exact n, df, tests, and $P$-values are provided in Table S2. "JC-1 red/green ratio (a.u.)" is used as the axis label in figures.

## TEM

Isolated mitochondria were fixed in 4% PFA + 2% glutaraldehyde in 0.1 M cacodylate buffer (5 mM CaCl$_2$, pH 7.4) for 24 h, postfixed in 1% osmium tetroxide with 0.5% potassium hexacyanoferrate (and potassium dichromate where noted) for 1 h, stained en bloc with 2% uranyl acetate (1 h), dehydrated through a graded ethanol series, and embedded in epoxy resin. Ultrathin sections (~70 nm; Leica EM UC7) were collected on 200-mesh copper grids, stained with lead citrate, and imaged on a Tecnai T12 Spirit (Thermo Fisher Scientific) equipped with a Gatan OneView camera.

## Oxygen consumption (Seahorse XF96)

Isolated mitochondria were assayed on an Agilent Seahorse XF96. On the day of the experiment, preparations were kept on ice in MAS (pH 7.2 at 37°C; see the Reagents and solutions section) + 0.2% fatty acid–free BSA. Protein concentration was determined (e.g., Bradford), and suspensions were first diluted 10× in cold 1× MAS containing the appropriate substrate(s), then adjusted to deliver 2.5 µg mitochondrial protein in 50 µl per well. Plates were kept on ice during loading. Mitochondria were adhered by centrifugation (2,000$g$, 10 min, 4°C; swinging-bucket adapter). Immediately afterward, 130 µl prewarmed (37°C) MAS + substrate was added to each well (to a final volume of 180 µl), and OCR was recorded at 37°C.

Complex I–linked substrate protocols include supplying MAS + malate 5 mM and glutamate 10 mM throughout the experiment and sequential injections (final) of ADP 4 mM, CCCP 4 µM (where applicable), rotenone 0.5 µM, and antimycin A 4 µM. Complex II–linked substrate protocols include supplying MAS + succinate 10 mM and rotenone 0.5 µM throughout the experiment and sequential injections (final) of ADP 4 mM, CCCP 4 µM (where applicable), and antimycin A 4 µM.

Background wells (buffer only) were included on each plate. Technical replicates per condition were averaged within each biological replicate, and OCR was normalized to mitochondrial protein loaded per well.

## Reagents and solutions

### Buffers

HIM (ice-cold): 200 mM mannitol (36.4 g; M4125; Sigma-Aldrich), 70 mM sucrose (23.95 g; S7903), 10 mM Hepes (2.38 g; H3375), 1 mM EGTA (0.38 g; E3889), 1 mM MgCl$_2$·6H$_2$O (M8266); pH 7.5 with KOH (P1767). MAS (for Seahorse): 220 mM mannitol (M4125), 70 mM sucrose (S7903), 10 mM KH$_2$PO$_4$ (P5655 or equivalent), 5 mM MgCl$_2$·6H$_2$O (M8266), 2 mM Hepes (H3375), 1 mM EGTA (E3889), 0.2% fatty acid–free BSA (A7030); pH 7.2 at 37°C. Cell culture medium (500 ml batch; final concentrations). Basal medium: DMEM, high glucose (4.5 g/liter), no pyruvate, 500 ml. FBS 50 ml (10% (vol/vol) final), L-glutamine (200 mM) or GlutaMAX: 5 ml (2 mM final), sodium pyruvate (100 mM) 5 ml (1 mM final), penicillin–streptomycin (100×) 5 ml (1× final) (100 U/ml penicillin, 100 µg/ml streptomycin), and Plasmocin prophylactic (InvivoGen) 50 µl per 500 ml (optional; used to maintain *Mycoplasma*-free cultures) were added under sterile and aseptic conditions, mixed gently to avoid foaming, kept sterile, and optionally passed through a 0.22-µm PES filter if needed. The mixture were stored at 4°C and used within 4 wk, and rewarmed to 37°C before feeding cells. Cell culture supplements used DMEM (high glucose, no pyruvate); FBS; L-glutamine 200 mM (or GlutaMAX); sodium pyruvate 100 mM; penicillin–streptomycin 100×; Plasmocin prophylactic (InvivoGen; 50 µl/500 ml, when indicated).

Poly-L-lysine (PLL) 0.01% (wt/vol) (P8920; Sigma-Aldrich) coating was used on AFM substrates.

Potentiometric/mitochondrial probes: TMRM (T668; Thermo Fisher Scientific/Invitrogen) (0.5 µM final) was used to stain mitochondria for 10–15 min on ice in HIM and quantified as mean pixel intensity (a.u.) in IDEAS. JC-1 (T3168; Thermo Fisher Scientific/Invitrogen) (5 µM final) was used to stain mitochondria for 10–15 min on ice in HIM, washed once by centrifuging at ~5,500$g$, 10 min, 4°C, and quantified as red/green ratio (F590/F530, a.u.) on a plate reader. MitoTracker Green FM (M7514; Thermo Fisher Scientific) (0.5 µM) was indicated for visualization. Respiratory substrates (working concentrations; non-Seahorse assays) used were as follows: L-malate 5 mM (M1000; or disodium L-malate M9138; Sigma-Aldrich), L-glutamate (monosodium) 10 mM (G5889; Sigma-Aldrich), succinate (disodium, hexahydrate) 10 mM (S2378; Sigma-Aldrich), TMPD (di-HCl) 0.5 mM (T3134; Sigma-Aldrich), and sodium ascorbate 10 mM (A4034; prepared fresh; Sigma-Aldrich). Inhibitors/modulators (working concentrations; non-Seahorse assays) used were as follows: ADP 200 µM (A2754; Sigma-Aldrich), rotenone (complex I inhibitor) 1.5 µM (R8875; Sigma-Aldrich), antimycin A (complex III inhibitor) 1 µM (A8674; Sigma-Aldrich), sodium azide (complex IV inhibitor) 5 mM (S2002; Sigma-Aldrich), CCCP (protonophore) 1 µM (C2759; Sigma-Aldrich), and valinomycin (K$^+$ ionophore) 1.8 µM (V0627; stock 1 mM in EtOH; carrier ≤ 0.1% vol/vol; vehicle-matched controls; Sigma-Aldrich). Seahorse XF96 injections (final concentrations in well, when applicable) used were ADP 4 mM, CCCP 4 µM, rotenone 0.5 µM, antimycin A 4 µM (exact injection order per figure legend).

### Carriers

DMSO/EtOH ≤ 0.1% (vol/vol) were used as carriers in all conditions; vehicles matched in controls.

### Figure preparation

Figure layout and panel assembly were performed for the following panels: Figs 1E, G–K, 2C–H, 3A–E, 4A–E, 5A–H, and S1F, G, I, K, and L. BioRender includes integrated AI-assisted design tools. Data plots were generated in (v9+) and exported for assembly; instrument-native software (IDEAS, JPK SPM, Bruker's NanoScope, OriginLab) was used where applicable. Axis labels and units were standardized (e.g., "PSD (a.u.)," "Integrated power 0–20 Hz [a.u.]," "JC-1 red/green ratio [a.u.]," "TMRM mean pixel intensity [a.u.]"). For PSD panels, mean spectra are shown; per-frequency SEM was below line width and is omitted for clarity, with inference reported on integrated power. Violin plots display full distributions with median and IQR. Colors were chosen for color-vision accessibility; fonts and line weights were unified across panels.

## Data Availability

This study includes no data deposited in external repositories. All data supporting the findings of this study are available within the article and its supplementary information files. Original data images are available from the corresponding author upon reasonable request.

## Supplementary Information

## Acknowledgements

We would like to express our deepest gratitude to the following individuals and groups for their invaluable contributions to this work. We thank Dr. Ziv Porat (Department of Chemical Research Support, Weizmann Institute of Science) for his assistance with flow cytometry (FACS). We are grateful to Dr. Elena Ainbinder and Dr. Kira Orlovsky (Seahorse Facility, Harry Levine Family Building, Weizmann Institute of Science) for their technical support with mitochondrial respiration assays. Special thanks to members of the Gross research group for their scientific guidance throughout this study. We also extend our heartfelt appreciation to Dr. Inna Grosheva for her valuable input and support. We acknowledge Dr. Nili Dezorella and Dr. Smadar Zaidman (Department of Chemical Research Support, Weizmann Institute of Science) for their expert assistance with TEM. Finally, we thank Dr. Melanie Bokstad Horev (Department of Biomolecular Sciences and Department of Immunology and Regenerative Biology, Imaging Center, Weizmann Institute of Science) for her support with imaging analyses.

### Author Contributions

EO Zorikova: formal analysis, investigation, and writing—original draft.
S Chourasia: investigation.
I Rosenhek-Goldian: investigation, formal analysis, and writing—review and editing.
SR Cohen: methodology, investigation, formal analysis, and writing—review and editing.
SV Nesterov: conceptualization.
A Gross: conceptualization, supervision, funding acquisition, and writing—review and editing.

### Conflict of Interest Statement

The authors declare that they have no conflict of interest.

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
