## [Reviewer comments · Life Science Alliance]

Bridging Nanomechanics and Bioenergetics of Single Mitochondria by Atomic Force Microscopy

Ekaterina Zorikova, Sabita Chourasia, Irit Rosenhek-Goldian, Sidney Cohen, Semen Nesterov, and Atan Gross
DOI: <https://doi.org/10.26508/lsa.202503602>

Corresponding author(s): Atan Gross, Weizmann Institute of Science

Review Timeline:	Submission Date:	2025-12-15
	Editorial Decision:	2026-01-27
	Revision Received:	2026-02-08
	Accepted:	2026-02-12

Scientific Editor: Sarita Hebbar

Transaction Report:

Please note that the manuscript was reviewed at *Review Commons* and these reports were taken into account in the decision-making process at *Life Science Alliance*.

Reviews

Review #1

****Summary:****

The authors use atomic force microscopy (AFM) to study mitochondria isolated from primary mouse livers, and they attempt to correlate these measurements with mitochondrial membrane potential and oxygen consumption under different bioenergetic conditions. They argue that AFM could be used diagnostically to assess mitochondrial function. While there is some novelty in potentially using AFM to assess mitochondrial function in the clinic, it is not clear how this would be more efficient or meaningful than assessing mitochondrial parameters by more standard methods, such as respirometry, confocal microscopy, etc. Considerably more work would need to be performed, particularly on relevant patient samples, to show that AFM holds potential as a diagnostic tool. It is important to note that the authors of this study have not taken sufficient care to quantify the mitochondrial membrane potential in a manner that could be considered reliable, which casts further doubt upon the merits of this method for diagnosing mitochondrial function. These concerns, laid out in detail below, should be thoroughly addressed before publication.

****Major comments:****

The authors used azide to inhibit complex V, but azide is also a potent inhibitor of complex IV (Bowler et al., 2006). Why did the authors not use oligomycin, which is more specific, to inhibit complex V?

In Fig. 1 H - K, the y axes are labelled in a confusing or ambiguous way. The legend says that all data represent the mean {plus minus} SEM; however, panels D, F, H, and K have no error bars. For example, the data in H and K are shown as violin plots. Typically, the y axis would say what the name of the quantity is (e.g., mean TMRM fluorescence intensity) followed by the units (e.g., a.u.) in parentheses. However, the authors write, for example, in panel K "Mean pixel (TMRM)." The authors seem to follow the correct convention in panels D - G, so it is not clear why H - K are written incorrectly. In any event, the authors need to specify how these data were obtained, as there are virtually no details as to the methods of how these measurements of mitochondrial membrane potential were acquired. For example, JC-1 is a ratiometric probe. In its monomeric form, it emits a green signal, but, as the dye aggregates into so-called J-aggregates, the emission is red. The correct way of analyzing JC-1 signal is to compute the ratio of red over green fluorescence intensity. However, in the authors' quantifications, they simply say "Fluorescence (JC-1)." The units of the y axes go from zero to 20,000, which means that the authors likely did not assess the ratio of these emissions, so the data are not informative as to the actual mitochondrial membrane potential. Moreover, the authors indicate that they use 5 μM JC-1. This seems quite a high concentration, particularly for staining isolated mitochondria, which means that the dye has direct access to the organelle without having to cross the plasma membrane. There is no information about how long the dye was allowed to load and whether it was washed off prior to obtaining the measurements with the plate reader. Likewise, the authors used TMRM to also try to assess the mitochondrial membrane potential. In this case, they used 0.5 μM , but they did not indicate for what duration the mitochondria were exposed to the dye before going through the FACS. It should be noted, too, that TMRM is a Nernstian probe, which effectively stains mitochondria at concentrations as low as 1 nM. Accordingly, it is known that TMRM (and other mitochondrial dyes) can be toxic at higher concentrations, inhibiting essential processes such as OXPHOS. The very low dynamic range of the TMRM signal in panels H and K suggest that the signal was saturated, because there was too much dye loaded into the mitochondria. Moreover, the values, ranging merely from zero to 80 suggest a very insensitive method for quantifying the mitochondrial membrane potential. In Fig. S1 A-B, the authors used confocal microscopy to assess the isolated mitochondria. It would be wise to continue to use this technique for the other experiments, as plate readers and FACS offer no direct visual cues to validate that the numbers reflect bona fide biological measurements. Especially in the case of FACS, where there is an exceedingly large number of events, the statistics become essentially meaningless, as it is possible to show that almost anything is statistically significantly different if there is a sufficiently high number of samples or events. The authors should bear in mind that measuring the mitochondrial membrane potential is not trivial. One needs to understand the properties of the probes that are being employed as well as the instruments that are used to make the measurements. Care must be taken to ascertain that the quantifications reflect true biological processes.

The authors claim, for Fig. 1, that there is an "excellent correlation" between height fluctuations and mitochondrial membrane potential. Given that the mitochondrial membrane potential measurements were associated with various errors (see above), it is premature to assert that there is any correlation, at all. Furthermore, if the authors want to argue that there is indeed a correlation between these variables, then they should perform an appropriate statistical analysis, e.g., a Pearson correlation coefficient test.

For the reasons explained above, the JC-1 and TMRM measurements in Figs. 3 and 4 are not convincing. The authors must demonstrate, unambiguously, that they understand the use of these probes and that they are making accurate measurements.

Given that MTCH2 was recently reported to function as an insertase of the OMM (Guna et al., 2022), understanding the KO phenotype is extremely challenging, since it implicates the downstream loss of function of numerous other proteins. It would be valuable to examine other KO models with more specific mitochondrial defects, which can simplify the interpretation of the data. For example, suppression of any of the large Dynamin GTPases that control mitochondrial shape, i.e., MFN1/2, OPA1, or DRP1. Conversely, modulation of mitochondrial membrane composition by suppression of specific phospholipid biosynthetic enzymes would be valuable. It is important to note that the authors are attempting to highlight AFM as a novel way to assess patient samples, but they do not provide any data as to whether mitochondria, derived from a patient with a known mitochondrial defect, could be meaningfully assessed by this method.

It is worth pointing out, too, that isolating mitochondria from primary tissues involves a significant amount of stress to the organelle. To understand mitochondrial function in a manner that reflects an *in vivo* state as much as possible, it would be essential to show that the isolated mitochondria from the liver are largely the same as those in intact liver cells. The authors should be aware that isolating live hepatocytes is far from a trivial thing to do (Charni-Natan & Goldstein, 2020). Simply mincing the liver and subjecting it to mechanical and enzymatic dissociation likely involves significant mitochondrial stress, which implies that the values derived from isolated mitochondria represent a highly non-physiological, even dysfunctional, condition. These are fundamental concerns which should be considered and discussed in any report that is lauding the potential diagnostic benefits of quantifying isolated mitochondria from primary tissues.

The authors say, in the discussion, "Accordingly, the AFM method employed here measured several characteristics such as morphology and elastic modulus of the structures, as well as fully exploiting the rich information available from the noise spectra." There was no measurement of "morphology" in this study. Differences in height are not what is generally considered in discussions of mitochondrial morphology, which reflects the dynamic changes in organelle shape and connectivity, typically in the x-y (rather than z) axes.

The authors performed experiments on fixed and dried mitochondria; however, there is no systematic comparison of the integrated power and other parameters compared to the live mitochondria isolates. This is a key comparison that should have been performed, as it would offer a basic frame of reference for the values of the live organelles. Another key experiment that is lacking in this study is measurement of the same organelle over time to understand the variance in individual organelles from moment to moment.

****Minor comments:****

Generally, the authors should moderate their claims that AFM could be used diagnostically until the above concerns are addressed.

There needs to be considerably more detail as to the methods that were used here. This is essential insofar as the authors wish to convince potential readers that the experiments were carefully conducted and that the data is reliable. Putting numbers on the margin of the manuscript would be helpful for the referee to specifically address certain points.

References:

Bowler MW, Montgomery MG, Leslie AG, Walker JE. How azide inhibits ATP hydrolysis by the F-ATPases. *Proc Natl Acad Sci U S A*. 2006 Jun 6;103(23):8646-9. doi: 10.1073/pnas.0602915103. Epub 2006 May 25. PMID: 16728506; PMCID: PMC1469772.

Guna A, Stevens TA, Inglis AJ, Replogle JM, Esantsi TK, Muthukumar G, Shaffer KCL, Wang ML, Pogson AN, Jones JJ, Lomenick B, Chou TF, Weissman JS, Voorhees RM. MTCH2 is a mitochondrial outer membrane protein insertase. *Science*. 2022 Oct 21;378(6617):317-322. doi: 10.1126/science.add1856. Epub 2022 Oct 20. PMID: 36264797; PMCID: PMC9674023.

Charni-Natan M, Goldstein I. Protocol for Primary Mouse Hepatocyte Isolation. *STAR Protoc*. 2020 Aug

I am an expert in imaging of mitochondria, with considerable direct knowledge of various super-resolution and advanced imaging systems. I have also studied mitochondrial function, using standard biochemical and molecular approaches. I have great familiarity with mitochondrial behavior and dynamics, as understood from live-cell imaging approaches and morphological analysis.

This study is potentially interesting due to its relatively novel use of AFM to examine mitochondria. However, there is a lot of uncertainty in the measurements due to technical oversights and lack of relevant controls. Whether AFM could be useful in the clinic remains an open question. If the authors could address the comments above, it would go a long way to finding out one way or the other.

Between 3 and 6 months

Yes

Review #2

In the article "Non-Invasive Mechanical-Functional Analysis of Individual Liver Mitochondria by Atomic Force Microscopy", O. Zorikova and colleagues propose the use of Atomic Force Microscopy (AFM) as a tool for characterizing the biophysical properties of individual mitochondria. By analyzing parameters such as height, membrane fluctuation power spectra, and Young's modulus under various drug treatments and genetic mutations, the authors aim to provide a novel, label-free method for assessing mitochondrial functionality.

While the manuscript presents an interesting approach, the introduction would benefit from a clearer and more cohesive narrative. The authors highlight the need to monitor the function of individual mitochondria, which is indeed an important challenge, but the rationale for doing so should be more explicitly stated. A stronger emphasis on the biological importance of mitochondrial biophysical parameters and the added value of using AFM would enhance the motivation for the study. Additionally, the symbol $\Delta\psi$, referring to mitochondrial membrane potential, should be defined and briefly explained in the introduction for clarity.

In the results section, a schematic diagram of the experiment would aid comprehension, especially for readers less familiar with this technique. In general, in the figures it would be good to find the individual data points. The integration of the results into the main text could also be improved. Currently, several findings are presented in a descriptive manner, but the biological interpretation or relevance is not always clear. For example, the sentence "Figure 2 presents a comprehensive analysis of the height and elastic properties of mitochondria" could be expanded to explain what those findings actually mean and how they help support the main goal of the study. Similarly, the statement that "the integrated power of mitochondrial membrane fluctuations decreased significantly upon valinomycin treatment" is presented without explanation of what this metric represents or why valinomycin was chosen. When discussing MTH2, the authors refer to "mechanical alterations in mitochondria lacking this protein" without explaining what MTH2 is, where it is localized, or why it is biologically relevant.

Finally, in the discussion, the interpretation of results could be expanded. For example, the statement "MKO/MLM exhibited increased integrated power/potential, increased modulus/stiffness, and decreased height" would benefit from more biological context - what do these changes imply about mitochondrial function or physiology? Adding this kind of interpretation would help the reader better understand the broader significance of the findings.

Methods: The authors say they record the piezo movement but it is not clear to the reviewer if the authors perform a closed-loop force-feedback experiment. If so, this will introduce noise into the measurement which can be avoided by performing an open loop measurement. Why did the authors not record the cantilever fluctuation at a constant piezo height? This gives enough bandwidth and low noise to record Angstrom deflections. Likewise, it is unclear to this reviewer why the power spectrum is given in V and not in nm, as it is typical in AFM measurements. I assume the authors calibrated the deflection sensitivity and spring constant of the cantilever, hence, if possible, the authors should convert the PSD into nm/Hz.

During the elasticity measurements, did the authors correct for the finite thickness of the mitochondria? What was the contact force and indentation depth, and how thick were the mitochondria to begin with? If the indentation is larger than 20%, I suggest to perform a correction to account for the infinite stiffness of the substrate. Given that the mitochondrial stiffness is in the tens of kPa, this seems to be important (perhaps not for relative values but for

absolute stiffness measurements).

Figures. The figures are well constructed and aid the reader through the important messages of the paper. The authors however, should not excessively overuse bar charts without explicitly mentioning number of measurements for each condition. In essence, I strongly recommend plotting individual data points to see the distribution and replace the stars with actual p-values.

The premise of the study is compelling and could have important clinical implications for distinguishing dysfunctional mitochondria in pathological contexts. However, the manuscript in its current should be improved. First of all, non-invasive is more than an euphemism, as the mitochondria need to be taken out of the cell, which is highly invasive. The authors should delete non-invasive from the title.

As the work presents an orthogonal and non-standard approach, the authors introduced a novel assay that can guide future investigations into the biophysics of mitochondrial physiology. Thus the paper is of high interest, timely and cutting edge.

In summary, the study presents a promising approach with potentially high relevance for mitochondrial research.

Between 1 and 3 months

No

Review #3

The article titled 'Non-Invasive Mechanical-Functional Analysis of Individual Liver Mitochondria by Atomic Force Microscopy' discusses how the mechanical properties of mitochondria as response to various drugs, like CCCP and ADP or rotenon and antimycin A.

Key findings:

The Authors correlated the thermal noise power spectrum (PSD) measured in contact on the top of mitochondria using atomic force microscope (AFM) with membrane activity of the organelles measured using fluorescence markers.

They identified correlation trends between PSD, height, elasticity and fluorescence marker intensities for various cases, where the organelle activity was modified using drugs or genetic changes.

The work is a very interesting approach, an excellent application of mechanobiology to gain further understanding of the properties of the energy producing organelles of eukaryotes.

However, the overall results the Authors present have some serious flaws.

I would recommend for publication after significant changes were made.

Major comments:

Upon measuring the power spectrum density (PSD) of thermal fluctuations in contact of an organelle, there are several factors influencing the measurements, such as: spatial inhomogeneity of the mitochondrion, the loading force applied, the feedback system of the AFM, hydrodynamic drag of the media on the cantilever.

None of the above points are addressed in the manuscript. That is:

- what was the spatial variability of the signal on the top of the organelle? (Using a tip with 30 nm apex radius has a relatively high variability even in microscopically homogeneous systems)
 - what was the loading force applied, and how did the PSD vary with the loading force?
 - according to the text on the bottom of page 5 the feedback was ON. How did this influence the recorded PSD?
- Significance of differences between organelles can be only properly estimated in relation to the spatial and load dependence of the same information.

Minor comments:

- Numerical Fourier transform generating the PSD is very noise prone, thus many curves need to be averaged for a

good result. Please provide statistical information on this aspect of the obtained curves.

- In the text it is mentioned that characteristic changes of the PSD were observed. What are the characteristic changes between unperturbed and drug affected mitochondria? Please highlight them on the graphs of PSD.
- How is the distribution of the results e.g. in Figure 1.E? Histogram and box-plots are more informative than bar plots.
- How many curves were recorded for the individual mitochondria? (30 mitochondria were measured)
- Figure 2.A and Figure S1.C indicate nicely how heterogeneous the mitochondria are. How did you eliminate the corresponding error from the PSD measurements?
- To highlight correlations, simple plots of the parameters as the function of each-other can be very informative.
- On Figure 1, the correlation between the fluorescence intensities and the PSD integrals are only qualitative.
- On Figure 3 the inverse correlation between the height and Young's modulus is not clear. Can it be plot such a way that the intended information becomes clear?
- While the Authors are claiming that the PSD is characteristic to the mechanical properties of the organelles, its direct connection remains elusive and is not discussed in the paper. Again, loading force dependence is expected to be present and influence whether the probe is detecting changes in membrane properties or sense something deeper, structures under the membrane.
- While the Authors correlate various measures derived from AFM data, these are only ensemble comparisons, since imaging and PSD measurements were done using different AFMs, thus different sample points. This should be clearly stated in the text.
- QI mode is very robust for imaging, but its Young's moduli are difficult to compare to any real situation, since the measurement is performed typically at the 500 - 2000 Hz frequency range. Not mentioning that the individual force curves are usually rather noisy for biological samples.
- In Figure S1.B, nothing is visible for the CCCP sample.
- In Figure S2, what does the value of 300 mean for alpha in the first sentence?
- While the frequency dependence of the PSD makes sense, the data indicated in figure S2 also indicates very high noise, making the fits unreliable. What would be the exponent value in the 5% - 95% confidence interval?
- It may be also informative to see a common plot of individual PSDs for the various cases, and in the representative plot see mean +/- SE plots for each frequency points.
- In the experiment description stands: 'Bruker Multimode AFM was used for overall imaging and power spectra in tapping mode.' This is misleading, because in tapping mode the end of the cantilever is driven by a constant frequency, which would interfere with the thermal PSD measurement. If it was done so, this is a driven state which should be discussed, and which is also dependent on the driving frequency.
- When preparing the PLL surfaces, how were the mica substrates washed before adding the organelles?
- The topography images are most probably measured Z-piezo sensor outputs. However, this is not mentioned.
- Imaging conditions of QI mode are incomplete the point measurement frequency, parameter to the apparent Young's modulus is not mentioned.

****Referee cross-commenting****

Reading the review of Reviewer 1 highlights the flaws in the organelle biology part of the work I was not aware of. (I am expert in mechanical characterization in the molecular - cellular level.) Putting the reviews together highlights that this study is in a very early state of investigation. It would be really interesting to see its results, but claiming it to be a novel diagnosis tool may be far fetched. (I agree with Referee 1.)

In general, the idea of estimating the mechanical properties of mitochondria and correlate them to the activity of the organelles is a very interesting idea in the field of mechanobiology.

The Authors have done a relatively large amount of experiments to identify correlation between activity followed by more traditional fluorescence labels and the AFM data they generated.

They performed many experiments spanning also three AFM devices and other experimental methods in their work.

Limitations:

I believe however, they missed some key points influencing their results, most importantly the dependence of the data on the:

- normal loading force
- spatial inhomogeneity (their own images prove the presence of this)

I am afraid some of the effects they detect are not only qualitative, but also biased, but with the current figures and data I cannot substantiate.

Audience: specific to microbiology, especially the audience interested in mechanobiology

I believe this is an interesting work, and contributes to our understanding of micromechanics at the organelle level. Thus I would really like to see it published in a more complete form.

Advance: Mitochondria is known to respond to environmental clues and can remodel its internal structure in response to stresses. However, it is difficult to find studies on the individual mechanical properties of these organelles, even in ex-situ environments.

Between 1 and 3 months

Yes

Manuscript number: RC-2025-03057

Corresponding author(s): Atan, Gross

[Please use this template only if the submitted manuscript should be considered by the affiliate journal as a full revision in response to the points raised by the reviewers.

*If you wish to submit a preliminary revision with a revision plan, please use our "Revision Plan" template. **It is important to use the appropriate template to clearly inform the editors of your intentions.**]*

1. General Statements [optional]

This section is optional. Insert here any general statements you wish to make about the goal of the study or about the reviews.

This section is mandatory. Please insert a point-by-point reply describing the revisions that were already carried out and included in the transferred manuscript.

Reviewer 1

R1.1

Why was azide used to inhibit complex V given that it also inhibits complex IV (Bowler et al., 2006)? Why not use oligomycin, which is more specific for complex V?

We used sodium azide to inhibit complex IV (cytochrome c oxidase)-our goal was to block electron transport at the terminal oxidase. This collapses the proton-motive force and thereby functionally abolishes ATP synthase activity, so adding oligomycin would not further change the intended bioenergetic state. We recognize that azide can inhibit F₁F₀-ATPase under some conditions¹, so we used doses/exposures where CIV inhibition predominates and confirmed CIV blockade (sharp decreases in oxygen consumption and $\Delta\Psi_m$). Consistently, oligomycin after azide had no additional effect. We clarify at the top of p. 7 that azide was employed as a complex IV inhibitor.

R1.2

a) In Fig. 1 H - K, the y axes are labelled in a confusing or ambiguous way. The legend says that all data represent the mean {plus minus} SEM; however, panels D, F, H, and K have no error bars. For example, the data in H and K are shown as violin plots. Typically, the y axis would say what the name of the

quantity is (e.g., mean TMRM fluorescence intensity) followed by the units (e.g., a.u.) in parentheses. However, the authors write, for example, in panel K "Mean pixel (TMRM)." The authors seem to follow the correct convention in panels D - G, so it is not clear why H - K are written incorrectly. In any event, the authors need to specify how these data were obtained, as there are virtually no details as to the methods of how these measurements of mitochondrial membrane potential were acquired. For example, JC-1 is a ratiometric probe. In its monomeric form, it emits a green signal, but, as the dye aggregates into so-called J-aggregates, the emission is red. The correct way of analyzing JC-1 signal is to compute the ratio of red over green fluorescence intensity. However, in the authors' quantifications, they simply say "Fluorescence (JC-1)." The units of the y axes go from zero to 20,000, which means that the authors likely did not assess the ratio of these emissions, so the data are not informative as to the actual mitochondrial membrane potential.

We revised the Y-axis labels to explicitly state the measured quantity and units and removed the ambiguous phrase "Mean pixel (TMRM)". Because these panels were acquired by imaging flow cytometry (Amnis ImageStream) and analyzed in IDEAS, the quantitative readout is the IDEAS feature Mean Pixel (average pixel intensity within the Object mask, with background subtraction). The Y-axis now reads "Mean Pixel intensity (TMRM, a.u.)" for Fig. 1H and 1K. For plate-reader panels we now use "JC-1 red/green ratio (a.u.)".

We now report exact p-values and n for all panels in a single supplementary file, Supplementary Table S1 (Statistical Summary for All Figures). For clarity, we kept significance asterisks on the plots; all corresponding exact p-values are provided in Supplementary Table S1.

Violin plots display the full distribution of per-mitochondrion measurements; overlaying SEM is not informative and reduces readability. Instead, we report per-mitochondrion n and exact p-values for each comparison. The center line denotes the median (inner box shows the IQR, when applicable). For 1H and 1K, n per group exceeds 1,000 mitochondria (totals ~4.5k and ~5.3k observations, respectively); one-way ANOVA with Tukey's multiple comparisons was used, and the exact p-values are provided in Supplementary Table S1.

All exact p-values (computed from F and degrees of freedom), n for every group, and the tests used (ANOVA/Tukey or t-tests) for every panel and figure are compiled in Supplementary Table S1.

We have now analyzed JC-1 ratiometrically (red/green) and added the full workflow to the Methods. See "Microplate fluorimetry (JC-1 $\Delta\Psi_m$, ratiometric)" section on page 21.

Figure axes were corrected to "JC-1 red/green ratio (a.u.)". The channel/filter settings, background subtraction, and ratiometric computation are now described explicitly in Methods.

b) Moreover, the authors indicate that they use 5 μM JC-1. This seems quite a high concentration, particularly for staining isolated mitochondria, which means that the dye has direct access to the organelle without having to cross the plasma membrane. There is no information about how long the dye was allowed to load and whether it was washed off prior to obtaining the measurements with the plate reader. Likewise, the authors used TMRM to also try to assess the mitochondrial membrane potential. In this case, they used 0.5 μM , but they did not indicate for what duration the mitochondria were exposed to the dye before going through the FACS. It should be noted, too, that TMRM is a Nernstian probe, which effectively stains mitochondria at concentrations as low as 1 nM. Accordingly, it is known that TMRM (and other mitochondrial dyes) can be toxic at higher concentrations, inhibiting essential processes such as OXPHOS. The very low dynamic range of the TMRM signal in panels H and K suggest that the signal was saturated, because there was too much dye loaded into the mitochondria. Moreover, the values, ranging merely from zero to 80 suggest a very insensitive method for quantifying the mitochondrial membrane potential.

Whereas we appreciate the reviewer's caution here, our choice of 5 μM lies well within the range used for isolated mitochondria in the literature and vendor protocols, which spans approximately 0.2–15 μM depending on preparation, readout, and desired S/N:

A peer-reviewed Bio-Protocol for mouse liver isolated mitochondria uses 15 μM JC-1 for 10 min (plate reader) and a post-stain wash, i.e., conditions that exceed our concentration².

Sigma's Isolated Mitochondria Staining Kit (JC-1-based) recommends 0.2 $\mu\text{g/mL}$ (~0.3 μM) for isolated mitochondria (microscopy/flow), illustrating that effective concentrations vary widely with assay geometry and instrument sensitivity.

Thermo Fisher's MitoProbe JC-1 guide emphasizes ratiometric red/green analysis and standard filter configurations; We revised our analysis following these recommendations³.

Because JC-1 is temperature-sensitive, we performed all steps on ice to stabilize the preparation during handling and keep conditions consistent across treatments; we confirmed expected pharmacologic responses (substrates \uparrow ratio; ADP modest \downarrow ; CCCP collapse), supporting that 5 μM did not saturate the signal under our conditions. (Low-temperature handling prior to acquisition is also used in flow-based $\Delta\Psi\text{m}$ assays to stabilize the signal)⁴.

In our revised version we added the exact concentration, wash, buffer, and on-ice handling to the JC-1 Methods section (p.21 «Microplate fluorimetry (JC-1 $\Delta\Psi\text{m}$, ratiometric»); re-analyzed and now report JC-1 strictly as red/green ratios with background subtraction (units: a.u.), and where noted, normalized to vehicle control = 1.

We used TMRM at 0.5 μM for isolated mitochondria because this concentration provided a robust $\Delta\Psi\text{m}$ -dependent dynamic range with high signal-to-noise in our preparation while preserving the expected pharmacological responses (substrate \uparrow , ADP modest \downarrow , CCCP collapse). The staining was brief ($\approx 10\text{--}15$ min, on ice, in HIM buffer, dark) followed by immediate ImageStream acquisition without fixation. We titrated 25 nM–1 μM and selected 0.5 μM as the best compromise between S/N and responsiveness. Importantly, no saturation was observed: distributions shifted appropriately across conditions, and the assay preserved sensitivity to both depolarization (CCCP) and partial polarization changes (ADP).

Use of 0.5 μM TMRM in isolated mitochondria is consistent with prior literature employing comparable concentrations in suspension assays (e.g., PLoS ONE 2012: “mitochondria were suspended in measurement medium supplemented with 0.5 μM TMRM...”, with expected changes upon energization/depolarization). This follows general guidance on TMRM as a potentiometric probe and mode-dependent considerations, in established reviews/protocols⁵.

We added TMRM units and scale meaning to Methods (Imaging flow cytometry (ImageStream; TMRM $\Delta\Psi\text{m}$) and standardized the Y-axis label to “TMRM mean pixel intensity (a.u.)” in panels 1H and 1K.

Despite the numerically modest scale, the assay was sensitive and responsive: energized states (substrates) shifted the distributions to higher Mean Pixel values, ADP produced a modest decrease, and CCCP collapsed the signal—exactly the expected $\Delta\Psi\text{m}$ -dependent responses for isolated mitochondria. These shifts were reproducible across $n = 3$ biological replicates and statistically significant (one-way ANOVA with post-hoc correction; exact p-values provided in Supplementary Table S1). To avoid large-N artifacts, all hypothesis tests were performed at the biological-replicate level; event-level histograms/violins are shown for transparency ($\sim 1,335$ events/condition pooled).

R1.3

In Fig. S1 A-B, the authors used confocal microscopy to assess the isolated mitochondria. It would be wise to continue to use this technique for the other experiments, as plate readers and FACS offer no direct visual cues to validate that the numbers reflect bona fide biological measurements. Especially in the case of FACS, where there is an exceedingly large number of events, the statistics become essentially meaningless, as it is possible to show that almost anything is statistically significantly different if there is a sufficiently high number of samples or events. The authors should bear in mind that measuring the mitochondrial membrane potential is not trivial. One needs to understand the properties of the probes that are being employed as well as the instruments that are used to make the measurements. Care must be taken to ascertain that the quantifications reflect true biological processes.

We used JC-1 for verification of all conditions, whereas the confocal measurements were done to show the two extremes - under active (malate + glutamate) and depolarized (CCCP) conditions, visually capturing the expected increase and collapse of $\Delta\Psi_m$. These measurements were performed under the same conditions used for the plate reader/FACS assays. Specifically, Fig. S1A–B in the Supporting Information shows TMRM-stained mitochondria under active (malate + glutamate) and depolarized (CCCP) conditions, visually capturing the expected increase and collapse of $\Delta\Psi_m$.

R1.4

The authors claim, for Fig. 1, that there is an "excellent correlation" between height fluctuations and mitochondrial membrane potential. Given that the mitochondrial membrane potential measurements were associated with various errors (see above), it is premature to assert that there is any correlation, at all. Furthermore, if the authors want to argue that there is indeed a correlation between these variables, then they should perform an appropriate statistical analysis, e.g., a pearson correlation coefficient test.

We thank the reviewer for requesting to quantify the correlation between these two techniques. We replaced qualitative terminology with formal statistics. Under matched bioenergetic conditions (NT, malate+glutamate, rotenone, CCCP), Integrated PSD changes correlate with $\Delta\Psi_m$ readouts. For TMRM (ImageStream, Mean Pixel) vs PSD, we obtained Pearson $r = 0.860$ (exact $p = 0.167$); Spearman $\rho = 1.00$ (exact $p \approx 0.083$); Kendall $\tau = 1.00$ (exact $p \approx 0.083$) with $n = 4$ conditions. Given the small n , we additionally report an exact permutation test ($p \approx 0.042$) supporting perfect monotonic ordering. We now report these coefficients and exact p -values in the Results (section «Mitochondrial height fluctuations measured by AFM correlate with $\Delta\Psi_m$ measured by potentiometric fluorescent dyes», pp. 5–6).” and compile all test details in Supplementary Table S1.

R1.5

For the reasons explained above, the JC-1 and TMRM measurements in Figs. 3 and 4 are not convincing. The authors must demonstrate, unambiguously, that they understand the use of these probes and that they are making accurate measurements.

To mollify these serious concerns, we reprocessed and now report the JC-1 data ratiometrically and clarified all TMRM parameters as follows:

JC-1 (plate reader). For all panels we now use the red/green ratio (590/530 nm) with channel-wise blank subtraction, calculated per well and then averaged per biological replicate. Y-axes have been updated to “JC-1 red/green ratio (a.u.)”. These changes are reflected in Fig. 1I, 1J, 3E, 5D, 5H and Supplementary Fig. S1K.

TMRM (imaging flow cytometry). Isolated mitochondria were stained with 0.5 μM TMRM for 10–15 min on ice (dark) in HIM buffer and acquired immediately on the ImageStream. Data are shown as event-level distributions; the quantitative readout is Mean Pixel intensity in the TMRM channel (IDEAS definition; a.u.). This concentration and workflow provided robust $\Delta\Psi\text{m}$ -dependent responses without detector saturation (substrates \uparrow , ADP modest \downarrow , CCCP collapse). Axis labels were standardized to “TMRM mean pixel intensity (a.u.)”. These changes are found into Fig. 1H, 1K and Supplementary Fig. S1L.

Exact acquisition/processing details are now provided in the Methods section (Microplate fluorimetry (JC-1 $\Delta\Psi\text{m}$ in isolated mitochondria) and Imaging flow cytometry (TMRM $\Delta\Psi\text{m}$)). Exact n, tests, and p-values for every panel are compiled in the new Supplementary Table S1 (Statistical Summary for All Figures).

After reprocessing/standardization, all conclusions in Figs. 3–4 are unchanged: energized conditions increase $\Delta\Psi\text{m}$ and AFM integrated power, inhibitors/uncoupler reduce both; genotype-dependent differences in Fig. 4 are preserved.

R1.6

a) *Given that MTCH2 was recently reported to function as an insertase of the OMM (Guna et al., 2022), understanding the KO phenotype is extremely challenging, since it implicates the downstream loss of function of numerous other proteins. It would be valuable to examine other KO models with more specific mitochondrial defects, which can simplify the interpretation of the data. For example, suppression of any of the large Dynamin GTPases that control mitochondrial shape, i.e., MFN1/2, OPA1, or DRP1. Conversely, modulation of mitochondrial membrane composition by suppression of specific phospholipid biosynthetic enzymes would be valuable. It is important to note that the authors are attempting to highlight AFM as a novel way to assess patient samples, but they do not provide any data as to whether mitochondria, derived from a patient with a known mitochondrial defect, could be meaningfully assessed by this method.*

b) *It is worth pointing out, too, that isolating mitochondria from primary tissues involves a significant amount of stress to the organelle. To understand mitochondrial function in a manner that reflects an in vivo state as much as possible, it would be essential to show that the isolated mitochondria from the liver are largely the same as those in intact liver cells. The authors should be aware that isolating live hepatocytes is far from a trivial thing to do (Charni-Natan & Goldstein, 2020). Simply mincing the liver and subjecting it to mechanical and enzymatic dissociation likely involves significant mitochondrial stress, which implies that the values derived from isolated mitochondria represent a highly non-*

physiological, even dysfunctional, condition. These are fundamental concerns which should be considered and discussed in any report that is lauding the potential diagnostic benefits of quantifying isolated mitochondria from primary tissues.

a) We thank the reviewer for this important suggestion and agree that testing additional genetic backgrounds strengthens the generality of our approach. We have performed additional experiments, and in the revised manuscript we added a new dataset (new Fig. 5) using mitochondria prepared from mouse embryonic fibroblasts (MEFs). We analyzed mitochondria from four MEF lines: WT, MTCH2 KO, MFN1 KO, and MFN2 KO. All measurements and statistics follow the same pipeline as in Fig. 1 (AFM integrated PSD DC–20 Hz; AFM height and apparent Young’s modulus; JC-1 red/green ratio).

Key results (Fig. 5; exact n, DF, tests, and p-values in Supplementary Table S1):

MFN1 KO and MFN2 KO show a marked reduction in $\Delta\Psi_m$ (JC-1 ratio) and AFM integrated power, together with stiffening (\uparrow Young’s modulus) and decreased height, relative to WT (all comparisons significant by one-way ANOVA/Tukey).

MTCH2 KO exhibits a distinct phenotype: higher $\Delta\Psi_m$ and higher AFM integrated power, with stiffening and reduced height compared to WT (unpaired two-tailed tests; $p < 0.001$). These results are very similar to the results we obtained with the MTCH2 knockout liver mitochondria (MKO) that appear in Figure 4.

These data demonstrate that our mechanical–functional readout on isolated mitochondria is also sensitive to mitochondria fusion defects (MFN1 KO and MFN2 KO), yielding genotype-specific “fingerprints.” Because our experiments are performed on isolated organelles (outside the network context), they are designed to be orthogonal to network-level dynamics per se.

We did not include OPA1/DRP1 or lipid-biosynthesis mutants in this revision to keep the scope manageable; systematic following the inner-membrane remodeling and fission pathways will be a logical next step.

Manuscript changes - New Fig. 5 (WT vs MTCH2 KO; WT vs MFN1 KO vs MFN2 KO) and legend. Methods updated to include MEF mitochondrial preparation and analysis settings. Supplementary Table S1: unified statistics for all figures, including Fig. 5.

Regarding applicability to real clinical cases: We agree that testing patient-derived material would be a logical next step, but it is beyond the scope of this proof-of-principle, methodological study on isolated mitochondria. Here we establish the AFM– $\Delta\Psi_m$ framework and its pharmacologic dynamic range. The assay is compatible with small input and standard isolation from primary tissues/cells, so translation to patient samples is technically straightforward. In the revision, we add a brief Discussion/Limitations note at the end of the Discussion section, stating this translational potential and that evaluation in

genetically/biochemically defined cohorts (e.g., OXPHOS defects) is planned as a separate follow-up study pending IRB approval and access to biobanked material.

b) Isolated mitochondria from rat and mouse livers are the most common models for mitochondrial research, and the correspondence of their functional parameters to those in hepatocyte cells has been repeatedly demonstrated⁶. Consensus on their suitability for research has existed in the bioenergetic community for more than half a century. The only limitation of isolated mitochondria compared to cell culture is the need to conduct studies in a short period of time (several hours) after their isolation. This condition was strictly preserved in our work. We also already document that isolated liver mitochondria preserve native structure and function using standard orthogonal QC:

1. Confocal TMRM shows polarized mitochondria with expected substrate-driven increase and CCCP collapse (Fig. S1A–B),
2. TEM demonstrates intact ultrastructure with well-defined cristae (Fig. S1C),
3. Seahorse OCR displays canonical activation by ADP/FCCP and inhibition by rotenone/antimycin A (Fig. S1F–G),
4. $\Delta\Psi_m$ by JC-1 (plate) and TMRM (FACS) mirrors these manipulations and aligns with AFM readouts (Fig. 1H–K, S1K–L).

These comparative structural (TEM), bioenergetic (OCR, $\Delta\Psi_m$), and mechanical (AFM) criteria indicate that our preparations remain in vivo-like physiology. A direct paired comparison of $\Delta\Psi_m$ /OCR/mechanics in intact hepatocytes vs. isolated mitochondria would present profound technical difficulties and is outside the scope of this organelle-centric method.

R1.7

The authors say, in the discussion, "Accordingly, the AFM method employed here measured several characteristics such as morphology and elastic modulus of the structures, as well as fully exploiting the rich information available from the noise spectra." There was no measurement of "morphology" in this study. Differences in height are not what is generally considered in discussions of mitochondrial morphology, which reflects the dynamic changes in organelle shape and connectivity, typically in the x-y (rather than z) axes.

We thank the reviewer for this comment. Indeed, morphological studies are a very minor part of this study. We now add an interpretation of Fig S1C, D morphology measurements in dry sample in the first paragraph of the discussion. We already show a representative image in Fig. 1A. However, more importantly, we change the sentence referred to here to state topography (height) instead of morphology. We have chosen to rely on the parameter of height (Fig. 2C, E, G and Fig. 3B), as this is most accurately

measured in AFM and not influenced by measuring parameters such as broadening by the tip. We measured both the actual structural/morphological changes by monitoring the height of the mitochondria by measuring the height of the mitochondria directly from AFM topography images. This data is presented in Fig. 2C, E, G and Fig. 3B.

R1.8

The authors performed experiments on fixed and dried mitochondria; however, there is no systematic comparison of the integrated power and other parameters compared to the live mitochondria isolates. This is a key comparison that should have been made, as it would offer a basic frame of reference for the values of the live organelles. Another key experiment that is lacking in this study is measurement of the same organelle over time to understand the variance in individual organelles from moment to moment.

We thank the reviewer for these suggestions.

The measurement under dried conditions was shown because it provides better resolution, allowing observations of the cristae structure. But it is impossible to compare this measurement with those done on viable mitochondria in fluid. The dried mitochondria are not viable and would not have any fluctuation related to activity. Not only is the environment different, but the mechanical properties of the tip, tapping frequency and amplitude and force constant are necessarily different when working on dried samples. However, we did perform the necessary control on a bare surface under buffer, without mitochondria (Fig. S1J) to make sure the fluctuations we measure are indeed due to live organelle activity.

The reason the measurements were limited to short time scales is due to unavoidable drift, so that the same position on mitochondria cannot be maintained for a long time. We would like to point out that the protocol was kept constant so that under each condition the organelles were measured under comparable stage.

R1.9 (Minor comments):

Generally, the authors should moderate their claims that AFM could be used diagnostically until the above concerns are addressed.

There needs to be considerably more detail as to the methods that were used here. This is essential insofar as the authors wish to convince potential readers that the experiments were carefully conducted and that the data is reliable.

Putting numbers on the margin of the manuscript would be helpful for the referee to specifically address certain points.

We agree and have moderated the text and expanded methodological detail as requested.

Scope/claims. We toned down language about clinical use throughout the Abstract and Discussion, presenting AFM as an orthogonal, label-free research assay and explicitly stating that patient-derived validation is future work. We removed forward-leaning phrasing (e.g., “diagnostic”) and limited claims to the demonstrated experimental scope.

Methods-added detail. We substantially expanded the Methods to fully describe assay setup and analysis steps so that the results are transparent and reproducible: (i) JC-1 is now analyzed strictly ratiometrically (Red/Green = F590/F530) with channel-wise background subtraction, filter sets, staining time, wash, handling on ice, and replicate structure; (ii) TMRM (imaging flow cytometry) is reported explicitly as IDEAS Mean Pixel (a.u.) with the object mask definition, gating/exclusion criteria, fixed detector settings, staining time, and replicate-level statistics; (iii) AFM acquisition clarifies liquid tapping, Z-piezo readout for fluctuation spectra, sampling/windowing for PSD integration (0–20 Hz), and elasticity mapping parameters (setpoint, indentation depth). For every figure we list n, the exact tests used, and exact p-values in new Supplementary Table S1.

References

1. Bowler, M. W., Montgomery, M. G., Leslie, A. G. W., & Walker, J. E. (2006). How azide inhibits ATP hydrolysis by the F-ATPases. *Proceedings of the National Academy of Sciences of the USA*, 103(24), 8646–8649. <https://doi.org/10.1073/pnas.0602915103>.
2. Renault, T. T., Luna-Vargas, M. P. A., & Chipuk, J. E. (2016). Mouse Liver Mitochondria Isolation, Size Fractionation, and Real-time MOMP Measurement. *Bio-protocol*, 6(15), e1892. <https://doi.org/10.21769/BioProtoc.1892>.
3. Molecular Probes (Invitrogen). (2006). *JC-1 and JC-9 Mitochondrial Potential Sensors* (Product Information Manual MP03168). Retrieved from Thermo Fisher Scientific.
4. Chen, G., Yang, Y., Xu, C., & Gao, S. (2018). A Flow Cytometry-based Assay for Measuring Mitochondrial Membrane Potential in Cardiac Myocytes After Hypoxia/Reoxygenation. *Journal of Visualized Experiments* (136), e57725. <https://doi.org/10.3791/57725>.
5. Scaduto, R. C., Jr., & Grotyohann, L. W. (1999). Measurement of mitochondrial membrane potential using fluorescent rhodamine derivatives. *Biophysical Journal*, 76(1 Pt 1), 469–477. [https://doi.org/10.1016/S0006-3495\(99\)77214-0](https://doi.org/10.1016/S0006-3495(99)77214-0).
6. Bustamante, E., Soper, J. W., & Pedersen, P. L. (1977). A high-yield preparative method for isolation of rat liver mitochondria. *Analytical Biochemistry*, 80(2), 401–408. [https://doi.org/10.1016/0003-2697\(77\)90661-3](https://doi.org/10.1016/0003-2697(77)90661-3)

Reviewer 2

R2.1

In the article "Non-Invasive Mechanical-Functional Analysis of Individual Liver Mitochondria by Atomic Force Microscopy", O. Zorikova and colleagues propose the use of Atomic Force Microscopy (AFM) as a

tool for characterizing the biophysical properties of individual mitochondria. By analyzing parameters such as height, membrane fluctuation power spectra, and Young's modulus under various drug treatments and genetic mutations, the authors aim to provide a novel, label-free method for assessing mitochondrial functionality.

While the manuscript presents an interesting approach, the introduction would benefit from a clearer and more cohesive narrative. The authors highlight the need to monitor the function of individual mitochondria, which is indeed an important challenge, but the rationale for doing so should be more explicitly stated. A stronger emphasis on the biological importance of mitochondrial biophysical parameters and the added value of using AFM would enhance the motivation for the study. Additionally, the symbol $\Delta\psi$, referring to mitochondrial membrane potential, should be defined and briefly explained in the introduction for clarity.

We agree and revised the Introduction to (i) Augment the motivation for single-mitochondrion analysis and (ii) articulate the added value of AFM relative to standard $\Delta\Psi_m$ dyes and OCR. Population-averaged methods can obscure inter-organelle heterogeneity, rare dysfunctional subpopulations, and rapid state transitions; analyzing isolated organelles targets the elementary bioenergetic unit while avoiding network-level coupling. We justify the added value of correlating size, stiffness, and activity at this size-regime.

R2.2

- a) In the results section, a schematic diagram of the experiment would aid comprehension, especially for readers less familiar with this technique. a) In general, in the figures it would be good to find the individual data points. The integration of the results into the main text could also be improved.*
- b) Currently, several findings are presented in a descriptive manner, but the biological interpretation or relevance is not always clear. For example, the sentence "Figure 2 presents a comprehensive analysis of the height and elastic properties of mitochondria" could be expanded to explain what those findings actually mean and how they help support the main goal of the study. Similarly, the statement that "the integrated power of mitochondrial membrane fluctuations decreased significantly upon valinomycin treatment" is presented without explanation of what this metric represents or why valinomycin was chosen.*

- c) *When discussing MTH2, the authors refer to "mechanical alterations in mitochondria lacking this protein" without explaining what MTH2 is, where it is localized, or why it is biologically relevant.*

We thank the reviewer for these suggestions.

- a) Schematic diagram: We now provide such a diagram in the graphical abstract.

We also note that in the current Ms., Figure 1, schemes are shown which depict the concept and the experimental flow.

- b) We have clarified this matter and changed the section title: "Mitochondrial elastic modulus and height fluctuations are inversely correlated" to "Comparison between elastic modulus and mitochondrial height across different bioenergetic manipulations" and emphasize we perform this measurement to explore the effect of height and modulus on the measured fluctuations, as well as to search for correlations between them.

As the reviewer requests, a more descriptive narrative of the biological interpretation now appears in the *Discussion* (paragraph beginning «Importantly, valinomycin experiments dissociate matrix volume changes from polarization», p. 13–14).

We also clarify in the text the goal and meaning of the experiments with valinomycin in the discussion section Changes in Integrated power are not due to changes in mitochondrial swelling.

- c) Mitochondrial carrier homolog 2 (MTCH2) is an outer-mitochondrial-membrane (OMM) protein that regulates mitochondria metabolism, dynamics, and apoptosis. More recently it was reported that MTCH2 might function as a protein insertase, shaping the OMM proteome¹, and/or a phospholipid scramblase, shaping the OMM phospholipid composition². In isolated mitochondria, altering MTCH2 is expected to change $\Delta\Psi_m$ -coupled nanomotions and surface mechanics by modifying membrane protein load/composition and contact-site organization-manifesting as coordinated shifts in our readouts (integrated fluctuation power \leftrightarrow $\Delta\Psi_m$, z-height, apparent Young's modulus) without invoking network-level morphology. We corrected the typo ("MTCH2", not "MTH2") and added a brief sentence in the Results (AFM analysis can detect functional defects in mitochondria) summarizing this rationale.

R2.4

Finally, in the discussion, the interpretation of results could be expanded. For example, the statement "MKO/MLM exhibited increased integrated power/potential, increased modulus/stiffness, and decreased height" would benefit from more biological context - what do these changes imply about mitochondrial

function or physiology? Adding this kind of interpretation would help the reader better understand the broader significance of the findings

We have expanded the discussion; section “AFM analysis can detect functional defects in mitochondria” to answer these points.

R2.5

Methods: a) The authors say they record the piezo movement, but it is not clear to the reviewer if the authors perform a closed-loop force-feedback experiment. If so, this will introduce noise into the measurement which can be avoided by performing an open loop measurement. Why did the authors not record the cantilever fluctuation at a constant piezo height? This gives enough bandwidth and low noise to record Angstrom deflections.

b) Likewise, it is unclear to this reviewer why the power spectrum is given in V and not in nm, as it is typical in AFM measurements. I assume the authors calibrated the deflection sensitivity and spring constant of the cantilever, hence, if possible, the authors should convert the PSD into nm/Hz.

We thank the reviewer for these comments.

a) Due to the delicate nature of the mitochondria, we worked in tapping, not contact mode. This means that the tip is constantly oscillating, and at larger amplitudes than the measured height fluctuations, so the signal of tip fluctuations cannot be used to represent the desired measurement. Instead, we measure the height modulation under feedback control (closed loop). To do this, we reduce the feedback gain to avoid any noise from the control system; while ensuring it is fast enough to observe the height changes with time (time constant of 2 ms or 500 Hz which is well beyond the bandwidth required to observe the fluctuations measured here of several tens of Hz).

These points are both very important, so we have now added clarification to the methods section to better describe our protocol.

To record the power spectra of height fluctuations, the AFM tip was placed on selected mitochondria without lateral scanning, and the z-piezo signal (height) (in V; approximate conversion 0.2 nm/mV) was routed to a Picoscope 3205F digital oscilloscope to acquire data for 10 seconds at a rate of 10,000 samples per second. The feedback setpoint (constant-amplitude tapping) was kept fixed.

b) As for the units, for the power vs. frequency plots it would be problematic to present the units since we have normalized the data to display the trends more clearly, as described in the methods section. There are two important observations we derive from this data – 1) The changes of the average PSD for different conditions, and 2) The overall shape of the curves – i.e, 1/f trend and characteristic peaks.

Neither of these latter require showing the numerical value of the y-axis since they relate to the shape of the curves, not their numerical value. However, we do agree that it is useful for the readers to have an idea of the size of fluctuations, so we give the conversion factor in of nm/V and indicate the typical size of fluctuations (on the order of several Ångstroms).

R2.6

During the elasticity measurements, did the authors correct for the finite thickness of the mitochondria? What was the contact force and indentation depth, and how thick were the mitochondria to begin with? If the indentation is larger than 20%, I suggest performing a correction to account for the infinite stiffness of the substrate. Given that mitochondrial stiffness is in the tens of kPa, this seems to be important (perhaps not for relative values but for absolute stiffness measurements).

The reviewer is correct that for thin samples/layers on a substrate, the substrate effect is traditionally considered to be relevant for depths larger than 10% of thickness. However, for a compliant material on a stiff substrate, as in our case, this effect is small, about 10% even for 30% penetration into the material, whereas the differences in modulus observed between the different conditions were about a factor of two. Here, the force applied was 120 pN and the indentation depth was not larger than 20% of the overall mitochondria height. (around 20-30 nm out of 200- 300 nm). So as the reviewer correctly states, the error here would be small, and since we are not concerned of absolute values, but of differences the important thing is to be certain that the substrate is not biasing our results.

To substantiate our claim, we note that a substrate effect would lead to an inverse relation between stiffness and height, but the results do not show this (compare, for instance, frames E and F in Fig. 5).

R2.7 *Figures: The figures are well constructed and aid the reader through the important messages of the paper. The authors, however, should not excessively overuse bar charts without explicitly mentioning number of measurements for each condition. In essence, I strongly recommend plotting individual data points to see the distribution and replace the stars with actual p-values.*

We have added individual data points in all cases where they would not detract from clarity.

We have maintained the asterisks in the graphs, but to comply with this suggestion for improved clarity we now provide supplementary Table 1 which compiles all the statistics in one easy to read place.

References

1. Guna A, Stevens TA, Inglis AJ, Replogle JM, Esantsi TK, Muthukumar G, et al. MTCH2 is a mitochondrial outer membrane protein insertase. *Science*. 2022;378(6617):317–322. <https://doi.org/10.1126/science.add1856>

- Li D, Rocha-Roa C, Schilling MA, Reinisch KM, Vanni S. Lipid scrambling is a general feature of protein insertases. *Proc Natl Acad Sci U S A*. 2024;121(17):e2319476121. <https://doi.org/10.1073/pnas.2319476121>

Reviewer 3

R3.1

Upon measuring the power spectrum density (PSD) of thermal fluctuations in contact with an organelle, there are several factors influencing the measurements, such as: spatial inhomogeneity of the mitochondrion, the loading force applied, the feedback system of the AFM, hydrodynamic drag of the media on the cantilever. None of the above points are addressed in the manuscript. That is: - what was the spatial variability of the signal on the top of the organelle? (Using a tip with 30 nm apex radius has a relatively high variability even in microscopically homogeneous systems)

The reviewer is correct that these are important considerations, and we add a sentence pointing out the measures we took: First, the buffer environment was constant, so hydrodynamic drag does not change from sample to sample. The significant differences between the samples also show that it is not a dominating effect relative to that of the mitochondrial activity which we measure. Secondly, we carefully controlled consistent tapping parameters for all measurements (in conditions that were also optimal for imaging in tapping mode) this enables us to be as gentle as possible for proper and non-invasive PSD measurement.

Each reported measurement consists of statistics on about 30 mitochondria so any variation in position, contact area, etc. reflect such variations.

To reduce positional variability, we always parked the tip at the apex/center of an isolated, well-adhered mitochondrion identified from the preceding topography frame; particles <~150 nm in height or touching neighbors were excluded. We analyzed n = 30 mitochondria/condition (N = 3), so any residual within-organelle spatial variability is reflected in the biological variance. Additive-only controls (PLL–mica without mitochondria) showed markedly lower PSD in the same band (Fig. S1J), supporting that the measured signal arises from organelles rather than the substrate.

R3.2

(what was the loading force applied, and how did the PSD vary with the loading force? - according to the text on the bottom of page 5 the feedback was ON. How did this influence the recorded PSD?)

We thank the reviewer for raising these issues.

Due to the delicate nature of the mitochondria, we needed to work in tapping, not contact mode. This means that the tip is constantly oscillating, and at larger amplitudes than the measured height fluctuations, so this signal cannot be used to represent the desired fluctuations. Instead, we measure the height modulation under feedback control (closed loop). To do this, we reduce the feedback gain to avoid any noise from the control system; while ensuring it is fast enough to observe the height changes with time (time constant of 2 ms or 500 Hz which is well beyond the bandwidth required to observe the fluctuations measured here of several tens of Hz).

These points are both very important, so we have now added clarification to the methods section to better describe our protocol.

Together, these controls (standardized positioning, fixed set-point, minimized/constant feedback, fixed hydrodynamics, substrate-only controls, and low-frequency analysis with detrending/normalization) bound the principal artefacts; the integrated 0–20 Hz power differences we report persist under these constraints.

R3.3 (Major)

Significance of differences between organelles can be only properly estimated in relation to the spatial and load dependence of the same information.

As mentioned above, we made sure to keep the tapping parameters constant and collect data always from the center of each mitochondrion.

While we agree that in the ideal case, we each experiment should track a single organelle in the same position on it and equivalent measurement conditions before and after adding the different additives. However, this is impossible since the tip needs to be retracted before adding new solutions, and all registry with previous measurement is then lost. Therefore, we rely on our protocol to measure multiple organelles to gain statistical significance. Several experiments were done for each condition, on different days and with different tips going through a cycle of additives seeing similar results statistically.

R3.4 (Minor)

Numerical Fourier transform generating the PSD is very noisy prone, thus many curves need to be averaged for a good result. Please provide statistical information on this aspect of the obtained curves

Each mitochondrion was recorded once for 10 s at 10 kHz sampling rate (one PSD per organelle). Traces showing drift or instability were immediately re-acquired at the same position under identical feedback settings. Thus, the dataset represents 30 independent mitochondria per condition (three biological replicates), each contributing one 10-s spectrum. We did not perform additional segmentation or

averaging of shorter windows within a single trace. The aim was to capture steady-state fluctuations under defined conditions rather than temporal evolution within one organelle.

R3.5 (Minor)

How is the distribution of the results e.g. in Figure 1.E? Histogram and box-plots are more informative than bar plots.

AFM metrics we use box-and-whisker plots with overlaid points (one point = one mitochondrion). Boxes show median and IQR; whiskers indicate the non-outlier range (Tukey); hollow circles mark outliers. For ImageStream (TMRM) we use violin plots with median and IQR overlays.

Sample sizes. AFM: $n = 30$ mitochondria per condition; $N = 3$ biological replicates. ImageStream (TMRM): $N = 3$; we display event-level distributions for transparency ($\approx 1,335$ events/condition pooled; ≈ 445 per replicate), but all inference is at the biological-replicate level. JC-1 (plate reader): $N = 3$ (typically 6 technical wells per replicate).

Axes (integrated PSD). For panels labeled “Integrated PSD (0–20 Hz, a.u.),” the y-axis is a single scalar per mitochondrion: the area under the PSD curve from 0 to 20 Hz (after normalization), computed by the trapezoidal rule,

$$\text{IntegratedPSD}_{0-20} = \sum_{i=1}^{N-1} \frac{1}{2} (\text{PSD}_i + \text{PSD}_{i+1}) (f_{i+1} - f_i)$$

“0–20 Hz” denotes integration bounds, not the axis range; the axis starts at 0 because the integrated power is non-negative.

No per-frequency SEM on PSD curves is given because at each frequency bin the SEM was negligible relative to the line width, so adding shading would only clutter the plots. Uncertainty and statistics are therefore reported for the integrated metric.

R3.6 (Minor)

How many curves were recorded for the individual mitochondria? (30 mitochondria were measured)

In our routine protocol we acquired one 10-s time series per mitochondrion and computed a single PSD from that record. Traces showing artifacts or non-stationarity were discarded and immediately re-acquired at the same tip position and load; thus, the effective count was 1 PSD per mitochondrion. To illustrate intra-organelle variability without reprocessing the entire dataset, we provide a representative antimycin-A example: two 10-s recordings at 10 kHz sampling rate, each partitioned into five non-overlapping 1-s windows (Welch/Hann), yielding 10 PSDs for that mitochondrion. The window-level coefficient of variation of the integrated PSD (0–20 Hz) was ~ 0.38 and ~ 0.26 in the two recordings, and ~ 0.305 overall

across the 10 window PSDs-i.e., ~30% within-mitochondrion variability. Group statistics in the manuscript are based on 30 mitochondria per condition from three independent experiments.

R3.7 (Minor)

Figure 2.A and Figure S1.C indicate nicely how heterogeneous mitochondria are. How did you eliminate the corresponding error from the PSD measurements?

We minimized the impact of spatial heterogeneity and contact geometry as follows. Before each PSD acquisition, we first acquired a tapping-mode topography map and then positioned the tip at (or near) the geometric apex of an isolated, round mitochondrion. We used identical substrate/buffer/preparation protocols and a fixed tapping set-point and gains within a run; objects showing aggregation, fusion, or abnormal shapes were excluded. To reduce geometry-dependent amplitude differences, PSDs were normalized to the mean amplitude in a featureless part of the spectrum before integration. PSD was analyzed per mitochondrion ($n = 30$ per condition; $n = 3$), and we compared groups at the population level. Importantly, the between-condition differences in integrated PSD persisted when restricting the analysis to mitochondria within comparable height and modulus ranges (central IQR), indicating that the effects are not explained solely by size/stiffness or tip-sample contact geometry.

R3.8 (Minor)

To highlight correlations, simple plots of the parameters as the function of each other can be very informative.

We now add table S3 which shows the observed changes in each measurement type according to environment.

R3.9 (Minor)

On Figure 1, the correlation between the fluorescence intensities and the PSD integrals are only qualitative.

We accept this criticism and now report quantitative correlations between integrated PSD (0–20 Hz) and $\Delta\Psi_m$ readouts under matched conditions (N/T, Mal + Glu, Rot, CCCP). In the legend Fig. 1 we provide: JC-1 ratio vs. integrated PSD: Pearson $r = 0.909$ ($n = 4$), exact permutation $p = 0.042$; Spearman $\rho = 1.00$ (exact $p = 0.083$); Kendall $\tau = 1.00$ (exact $p = 0.083$).

TMRM Mean Pixel vs. integrated PSD: Pearson $r = 0.860$, exact $p = 0.167$; Spearman $\rho = 1.00$; Kendall $\tau = 1.00$.

The small n reflects the four matched conditions; all metrics show the same monotonic ordering, supporting a strong PSD– $\Delta\Psi_m$ association.

We also confirmed that the correlation holds when considering the extent of change (Δ) across the four pharmacological conditions—that is, the relative change in both PSD and $\Delta\Psi_m$ with respect to the non-treated control. The rank ordering of these Δ -values remains identical, further supporting a consistent relationship between spectral power and membrane potential.

Exact values and statistical tests are provided in Supplementary Table S1.

R3.10 (Minor)

On Figure 3 the inverse correlation between the height and Young's modulus is not clear. Can it be plotting in such a way that the intended information becomes clear?

Our goal in Fig. 3 was not to claim a universal inverse relationship between height and stiffness, but to show that the AFM fluctuation metric (integrated PSD) tracks $\Delta\Psi_m$ independently of geometric changes. Under valinomycin, the matrix swells and mitochondria become taller while the apparent Young's modulus decreases (softening). In contrast, depolarization/inhibition (e.g., CCCP, rotenone, antimycin A, azide) yields reduced height together with increased stiffness. We now state this explicitly in the Results and figure 3. legend and kept the same group order across the height and modulus panels to facilitate side-by-side comparison. The distributions are shown as measured; no replotting was required to support the conclusion that integrated PSD follows $\Delta\Psi_m$ even when height increases (valinomycin) while stiffness decreases.

R3.11 (Minor)

While the Authors are claiming that the PSD is characteristic to the mechanical properties of the organelles, its direct connection remains elusive and is not discussed in the paper. Again, loading force dependence is expected to be present and influence whether the probe is detecting changes in membrane properties or sense something deeper, structures under the membrane.

In general, noise is related to energy dissipation. In liquid tapping mode we record the Z-piezo displacement, i.e., height fluctuations at the geometric apex of an individual mitochondrion. The low-frequency PSD (0–20 Hz) captures a combination of: (i) active, $\Delta\Psi_m$ -dependent processes (ion/water fluxes and membrane-tension modulations), (ii) passive thermal fluctuations shaped by the organelle's viscoelastic properties, and (iii) small instrumental contributions, which we minimize by detrending and by normalizing to a featureless band.

We consistently find that the integrated PSD power varies monotonically with $\Delta\Psi_m$, whereas the apparent Young's modulus can change in the opposite direction (e.g., with valinomycin: height \uparrow , modulus \downarrow , while PSD follows $\Delta\Psi_m$). This indicates that integrated PSD primarily reflects energy-dependent biomechanical activity, which can be modulated but is not determined by elastic properties.

Load/geometry sensitivity. To avoid artifacts, we used identical buffer/substrate across groups (constant hydrodynamic drag), positioned the tip at the apex based on prior topography, and kept the set-point and feedback gains identical within a run, using gentle contact. Objects with aggregation/abnormal shape were excluded. A bare PLL–mica control showed a flat PSD (Fig. S1J).

During the tapping mode measurements, we used the maximal setpoint ratio allowing stable contact, i.e., minimal applied force.

We have added these clarifications to the Discussion (“What the PSD measures and its relation to mechanics”) and briefly to Methods (AFM-Fluctuation time series and PSD). This point is elaborated in the first two paragraphs of the Discussion (pp. 12–13), where we explain what the PSD represents and how it relates to mitochondrial mechanics.

R3.12 (Minor)

While the Authors correlate various measures derived from AFM data, these are only ensemble comparisons, since imaging and PSD measurements were done using different AFMs, thus different sample points. This should be clearly stated in the text.

We agree and now state this explicitly. PSD time-series (liquid tapping; Z-piezo readout) were recorded on a Bruker MultiMode, whereas quantitative elasticity maps (QI force mapping) and height were acquired on a JPK NanoWizard III. We chose distinct instruments/operating modes because each is optimal for the respective measurements (stable tapping with external analog output for PSD; high-throughput force mapping with precise load control for modulus). All measurements used the same mitochondrial preparations, buffer, and PLL-mica substrate, performed on the same day with same mitochondrial preparations and additives within a ≤ 3 -hour window under identical environmental conditions. This also allowed us to complete an entire series of measurements with height, fluctuation, and modulus measurements within the < 3 -hour time window to ensure viability of mitochondria. A limitation of this design is that PSD and elasticity cannot be paired one-to-one on the same individual organelle as different samples (although from same preparation on same day) were measured by the different instruments. Therefore, correlations are made at the population level. We now acknowledge this explicitly in Methods and Discussion.

R3.13 (Minor)

QI mode is very robust for imaging, but its Young's moduli are difficult to compare to any real situation, since the measurement is performed typically at the 500–2000 Hz frequency range. Not mentioning that the individual force curves are usually rather noisy for biological samples.

We have added a representative raw force–distance curve to the SI (now Fig. S3) and expanded Methods with the exact QI settings and curve-quality criteria. In brief:

Acquisition (QI, JPK NanoWizard III). Maps $5 \times 5 \mu\text{m}^2$, 128×128 px; approach speed $\sim 30 \mu\text{m}\cdot\text{s}^{-1}$, retract $\sim 50 \mu\text{m}\cdot\text{s}^{-1}$; Z-range $\sim 1 \mu\text{m}$; trigger force 120–150 pN. These settings yield a per-pixel cycle time ≈ 40 –60 ms (effective point rate ~ 20 –25 Hz), well below the cantilever resonance, so the modulus reflects a quasi-static response.

Curve quality / inclusion. Only curves with (i) a straight, stable pre-contact baseline, (ii) a clear contact point, (iii) indentation $< \sim 20\%$ of local height (to minimize substrate effects), and (iv) Hertz-fit $R^2 \geq 0.95$ were analyzed. Curves showing baseline drift, excessive hysteresis, or poor fits were discarded.

Apparent Young's modulus was obtained by Hertz fitting (Poisson's ratio = 0.5; spherical tip radius ~ 30 nm). Per-mitochondrion values were averaged from a central ROI to avoid edge effects.

R3.14 (Minor)

In Figure S1.B, nothing is visible for the CCCP sample.

This is expected for TMRM, a strictly $\Delta\Psi\text{m}$ -dependent, non-fixable live dye. In Fig. S1A–B we acquired images with identical settings: with malate+glutamate punctate mitochondrial TMRM signal is evident (S1A), whereas after CCCP the $\Delta\Psi\text{m}$ collapses and TMRM rapidly redistributes to the medium, so the mitochondrial puncta disappear (S1B).

R3.15 (Minor)

In Figure S2, what does the value of 300 mean for alpha in the first sentence?

n is the number of data points. This information now appears in the related Table S2.

R3.16 (Minor)

While the frequency dependence of the PSD makes sense, the data indicated in figure S2 also indicates very high noise, making the fits unreliable. What would be the exponent value in the 5% – 95% confidence interval?

This is indeed an important detail. We now quantify uncertainty for the log–log slope (α) and for goodness-of-fit. We see that there is difference of at least 2-3 standard deviations between untreated or stimulated mitochondria and the controls or inhibited ones.

We thank the reviewer for this request. We now provide the standard deviation derived from making a least-squares fit to the log-log plots.

R3.17

It may be also informative to see a common plot of individual PSDs for the various cases, and in the representative plot see mean \pm SE plots for each frequency point.

We appreciate the suggestion. In our dataset, overlaying individual PSDs ($n = 30$ independent measurements per condition, 1-Hz bins from 0–70 Hz) produces visually dense panels that obscure between-condition differences. Our prespecified statistical endpoint is the integrated low-frequency power (DC–20 Hz), which captures the biologically relevant changes and underlies all inferences.

To address transparency, we (i) keep the mean spectra for qualitative context, (ii) report quantitative summaries via the integrated DC–20 Hz power distributions and the linear fit, reporting both the fitted slope (α) and uncertainty (Fig. S2; Methods), and (iii) provide all individual PSDs and the exact FFT/integration pipeline as machine-readable Source Data so readers can reproduce any desired overlays.

R3.18 (Minor)

In the experiment description stands: “Bruker Multimode AFM was used for overall imaging and power spectra in tapping mode.” This is misleading, because in tapping mode the end of the cantilever is driven by a constant frequency, which would interfere with the thermal PSD measurement. If it was done so, this is a driven state which should be discussed, and which is also dependent on the driving frequency.

During PSD recordings the lateral scan was halted and the AFM operated in liquid amplitude-modulation (tapping) only to maintain gentle intermittent contact. For the PSD measurement, we used the Z-piezo displacement (height-feedback) signal under closed loop, i.e., the controller output that compensates for nanometer-scale height changes of a single mitochondrion. The appropriateness of this time response is discussed above (R3.1). The tapping drive is ~ 15 kHz, three orders of magnitude above the analyzed band (up to 70 Hz), and should not couple to the low frequency motions that we analyze. Feedback gains were minimized (constant setpoint) to avoid injecting control noise. Controls on bare PLL–mica yielded flat/low PSD, confirming that the measured power arises from organelle fluctuations rather than the drive. In response to this and comments of other reviewers, this is now detailed in the experimental section.

R3.19 (Minor)

When preparing the PLL surfaces, how were the mica substrates washed before adding the organelles?

After PLL incubation we never let the surface dry. The PLL solution was aspirated, and the mica was rinsed 3× with ddH₂O (~1 mL per rinse, gentle flow across the surface, tilting to drain), followed by a final rinse with ice-cold HIM (~1 mL). The substrate was kept continuously wet in HIM and mitochondria in BSA-free HIM were deposited within ~5–10 min.

Methods (add under “AFM sample preparation (PLL–mica)”)

Rinsing/wash sequence. After coating freshly cleaved mica with 0.01% (w/v) PLL for ≥30 min at RT, the PLL solution was aspirated and the substrate was rinsed three times with ddH₂O (~1 mL per rinse for 12 mm disks; gentle pipetting across the surface, tilt to drain) to remove unbound polymer, then once with ice-cold HIM (~1 mL) to equilibrate ionic strength and pH. The surface was always submerged in HIM (no drying). Mitochondria suspended in BSA-free HIM were applied immediately after the final HIM rinse, incubated 10 min on ice to adhere, then gently washed 3–4× with 200 μL HIM to remove non-adherent particles.

R3.20 (Minor)

The topography images are most probably measured Z-piezo sensor outputs. However, this is not mentioned.

The topographies shown in Figures 1A and SI 2 were performed in our Bruker MMAFM which does not have z-piezo sensors but rather uses the calibrated height (nm per V) values in the system parameters.

R3.21 (Minor)

Imaging conditions of QI mode are incomplete - the point measurement frequency, parameter to the apparent Young's modulus is not mentioned.

Added to methods section:

“Force curves from the center of each mitochondrion were used to calculate the elastic modulus, by applying contact mechanics Hertzian model (using JPK data processing software version 6.1.86). With Poisson ratio of 0.5 and spherical tip shape with 30 nm radius. The maximum force applied on each pixel was 120-140 pN and the approach speed was 30 μm/s.”

January 27, 2026

RE: Life Science Alliance Manuscript #LSA-2025-03602

Atan Gross
Weizmann Inst. of Science
Biological Regulation Department
The Weizmann Institute of Science
Candiotty Bldg, Rm 306a
Rehovot, IL-Rehovot 76100 76100
Israel

Dear Dr. Gross,

Thank you for submitting your revised manuscript entitled "Bridging Nanomechanics and Bioenergetics of Single Mitochondria by Atomic Force Microscopy".

Your manuscript was carefully evaluated by all the original reviewers. As you will note, your revisions addressed most of their concerns. However the reviewers have suggested some (textual) changes to strengthen this manuscript. Adding new experimental evidence is not necessary but we agree that you must address all the points raised by the reviewers with textual edits to the manuscript. These points encompass:

- accurately describing results and methods as indicated by Reviewers 1 and 2.
- moderating language to tone down claims/conclusions and extending the discussion to elaborate on limitations of investigating the structure-function relationship of mitochondria in ex-vivo conditions (Reviewer 1).
- Including missing citations, correcting presentation of results in text and figures, and eliminating redundancies in descriptions (all reviewers).

Overall, we would be happy to publish your paper in Life Science Alliance pending resolution of the above points and final revisions necessary to meet our formatting guidelines. We request you to submit a revised manuscript document with all these changes highlighted, along with a point-by-point response to the reviewers comments.

MANUSCRIPT ORGANIZATION AND FORMATTING:

To avoid unnecessary delays in the acceptance and publication of your paper, please read the following information carefully. Full guidelines are available on our Instructions for Authors page, <https://www.life-science-alliance.org/authors>
****Submission of a paper that does not conform to Life Science Alliance guidelines will delay the acceptance of your manuscript.****

- Thank you for providing a statement on ethics compliance for animal work. Please provide the name of the institution approving the ethics protocols.
- Please transfer the HIM buffer composition to the section on reagents and solutions. Please cite this section when referring to a reagent/solution for the first time in your methods and do provide a pH for the buffers.
- When acknowledging the use of BioRender for the preparation of figures, please refer to the specific image panels ,and if applicable, kindly indicate that BioRender utilises an integrated AI-assisted design model.
- In the figure legends, please define N/T.
- Please provide details on MEF source and preparation. Please also include details on preparation of MTCH2 KO, MFN1 KO or MFN2 KO or provide a relevant citation for each.
- Kindly specify the samples for the section on 'Fixed/dried reference imaging'.
- Please remove the section summarising availability of supplemental information (lines 612-622) and kindly ensure that all required information from this section is included in the provided table titles or figure titles or figure legends.
- Please upload all figure files as individual ones, including the supplementary figure files; all figure legends should only appear in the main manuscript file, and remove the Supporting Information file.
- Please add a Running Title and a Summary Blurb/Alternate Abstract in our system.
- Please add a Category for your manuscript in our system.
- Please add the X and Bluesky handles of your host institute/organisation, as well as your own and/or one of the authors, in our system.
- Abstract should be a single paragraph not exceeding 175 words.
- Please incorporate any points from the Conclusion section into the Discussion; we only allow a Discussion section.
- Please add your main, supplementary figure, and table legends to the main manuscript text after the references section.
- The "Data Availability" section should be placed after the Materials & Methods section. Please consult our guidelines at

<https://www.life-science-alliance.org/manuscript-prep#format>

-Please add an Author Contributions section to your main manuscript text and the system.

-Please add a Conflict of Interest statement to your main manuscript text.

-Please be sure that the authorship listing and order is correct.

LSA encourages authors to provide a 30-60 second video where the study is briefly explained. We will use these videos on social media to promote the published paper and the presenting author (for examples, see <https://docs.google.com/document/d/1-UWCfbE4pGcDdcgzcmiuJl2XMBJnxKYeqRvLLrLS08s/edit?usp=sharing>). Corresponding or first-authors are welcome to submit the video. Please submit only one video per manuscript. The video can be emailed to contact@life-science-alliance.org

FINAL FILES:

The following items are required for acceptance.

Thank you for your attention to these final processing requirements. Please revise and format the manuscript and upload materials as soon as you are able.

Thank you for this interesting contribution to the literature. We look forward to publishing your paper in Life Science Alliance.

Sincerely,

Sarita Hebbar, PhD
Scientific Editor
Life Science Alliance
<http://www.lsjournal.org>

Reviewer #1 (Comments to the Authors (Required)):

Zorikova et al. use atomic force microscopy (AFM) to endeavor to correlate different topographical or physical properties of

mitochondria with specific functional readouts. They provide genetic and pharmacological evidence that depolarized mitochondria have a lower integrated PSD, whereas increased polarization leads to higher integrated PSD. The relationship between mitochondrial membrane potential and other AFM readouts, e.g., height and Young's modulus, appear inconsistent, however. The authors propose that AFM could provide a method for obtaining mitochondrial structure-function information without the need for dyes or other fluorescent labels. Although the authors provide evidence of a correlation between $\Delta\Psi_m$ and integrated PSD, specifically, the usefulness, and indeed the physiological relevance, of their approach comes with important caveats, which would need to be addressed prior to publication.

Major comments:

Lines 42-44: The authors state, "Under various respiratory manipulations, a simple scalar of low frequency height fluctuations closely paralleled the known signal to be unrelated to matrix swelling." This sentence is quite unclear. It is strongly recommended to revise it. For instance, "the known signal to be unrelated" requires clarity and context.

Lines 119-120: "The height measurement can serve as a proxy for volume/osmotic state and cristae packing." This sentence has no citation. There appears to be no evidence that height measurement corresponds to "cristae packing." Whether a "tall" mitochondrion has more or fewer or the same number of cristae (per cubic micron) as a "short" mitochondrion is unknown. Thus, this sentence should be revised or removed.

Lines 146-149: As a proof that the mouse liver mitochondria (MLM) are purified and functional, the authors provide confocal images of isolated MLM stained with the potentiometric dye TMRM. To demonstrate that these mitochondria harbor a membrane potential, they add the uncoupler CCCP, which should dissipate the TMRM due to depolarization. Figures S1A and S1B require clarification. Firstly, can the authors confirm that a window/level function was applied to S1A and S1B such that the contrast is identical between the two representative images? The signal in S1A is so faint that it is difficult to discern the difference between it and S1B without zooming in a lot. Notably, the brightest puncta, which are presumably isolated organelles, appear to be as large as 5 or 6 microns in diameter. The EM image in S1C, by comparison, shows a field of isolated mitochondria with diameters ranging from 0.5 to 1 micron, as expected. Although the images in S1A and S1B are obtained with confocal microscopy and S1C was taken with EM, the difference in size between the isolated organelles in these images should not be so different (e.g., as much as a factor of 10). The diffraction limit associated with confocal microscopy (on the X-Y axis) is around 200 to 250 nm, thus the considerable size of many of the bright puncta in S1A cannot be due to limited resolution. How can the authors explain this discrepancy? Why show a huge field of view with a 50-micron scale bar? This way of displaying this important proof is not actually providing persuasive evidence. At a minimum, there should be zoomed-in images that clearly highlight the differences, which the authors argue exist.

Lines 153-155: "High-resolution AFM of chemically fixed/dried mitochondria provided surface topography for morphology verification (Fig. S1D, E). These images give higher resolution, revealing the cristae folds." On the contrary: these images do not provide evidence of "cristae folds." As the authors will note from S1C, cristae are infoldings of the IMM; the images in S1D and S1E obviously do not capture these ultrastructural details. Thus, this language should be removed.

Lines 194-196: "These results demonstrate that AFM-based assessment of integrated power provides a reliable and comparable measure of $\Delta\Psi_m$, on par with established fluorescence based techniques." The authors must note here that AFM-based assessment of $\Delta\Psi_m$ is indirect and is, by its very nature, correlative. For this reason, it cannot possibly be "on par" with imaging approaches that directly measure the relative proportions of Nernstian dyes on either side of a membrane, which can, in turn, be used to compute absolute voltages of whole organelles or even individual suborganellar membrane potentials. The authors need to moderate this language so as to not overstate the power of their approach.

Lines 203-205: "By imaging a field of view containing multiple mitochondria, as shown in Figure 1A, AFM allows one to measure height with nanometer precision." Again, the authors appear to overstate the power of this approach. The mitochondria in Figure 1A, one or two of which is about a micron in diameter, show no signs of submicron detail, which would, of course, correspond to the existence of cristae, which are evident in either EM or super-resolution imaging approaches. The authors must be careful to refrain from making claims that are self-evidently not supported by their data.

Lines 232-233: "It is known from transmission electron microscopy experiments that an increase in $\Delta\Psi_m$ can cause swelling of the mitochondrial matrix." This sentence is missing a citation. Furthermore, the construction of this sentence should be improved. EM cannot provide a readout of $\Delta\Psi_m$, because the mitochondria are fixed or frozen. The classic Hackenbrock studies provide correlative information about the ultrastructure of "condensed" or "orthodox" mitochondria and their bioenergetics. But this is correlative. This fact should come across more clearly in the language here. This statement is further complicated by the fact that super-resolution imaging shows that depolarization of mitochondria with the uncoupler FCCP induces dramatic swelling of the mitochondrial matrix. Thus, while the authors state that it is known that an increase $\Delta\Psi_m$ causes swelling, it is also evident that a diminished $\Delta\Psi_m$ occurs upon depolarization. Of course, this complicates the picture substantially.

Lines 234-263: The authors go on for some time about effects of valinomycin and $\Delta\Psi_m$. While they indicate that valinomycin is known to dissipate $\Delta\Psi_m$ in the literature, whether valinomycin has the same effect in their hands is unclear—namely, in Figure 3D, the violin plot purports to show a significant change in $\Delta\Psi_m$ between "NT" and "Val," but it is not clear if the change is up or down. Indeed the nature of the violin plot makes it extremely difficult, at least in this case, to see where the mean would lie,

exactly. Please indicate what the mean is numerically in either condition to clarify whether the difference is an increase or decrease upon valinomycin treatment. Concordantly, the data in Figure 3E indicate that there is no change in $\Delta\Psi_m$. Thus, the overall conclusions drawn as to the effects of valinomycin on the $\Delta\Psi_m$ and the relationship to the AFM parameters in Figures 3A-3C are ambiguous. By contrast, CCCP has a clear effect on membrane potential in both the TMRM- and JC-1-stained mitochondria, so any conclusion about the relationship between $\Delta\Psi_m$ and the correlated AFM parameters would be more tenable with respect to this treatment. The authors' statement that "These structural and mechanical changes were accompanied by a loss of $\Delta\Psi_m$, confirmed by reduced TMRM mean pixel intensity (Fig. 3D) and lower JC-1 red/green ratio (Fig. 3E)." is not actually borne out by their data-particularly in the case of Fig. 3E, which shows no significant difference. These data must be revisited, as the specific conclusions appear to be unwarranted. If the authors wish to keep the valinomycin data, it would be recommended to perform a dose-response to determine a concentration that induces clear depolarization (shown by both loss of TMRM and a decrease in the red/green ratio of JC-1; conversely, this ambiguous data could be removed, as it does not appear to add much value to the overall picture).

Lines 1, 41, 287-289, 308, 407, etc.: References to "nanomotions" and "nanomechanics" and "nanoscale" should be removed, as the provided AFM images do not reflect the measurement of nm-resolution structures in this study.

Figures 3-5 are all missing representative images of the AFM. It is difficult to assess the effects of the pharmacological and genetic perturbations without being able to examine the actual images themselves. The authors must include representative images in these figures.

Lines 343-360: The discussion goes into some detail about how this study showed new information about MFN2 and the coenzyme Q10 pool. Which figures or panels reflect this narrative?

Lines 368-372: As indicated above, the valinomycin data is inconclusive.

Given the emphasis of this study on examining the topography of mitochondria using AFM on isolated mitochondria, it would be appropriate to discuss the limitations of investigating the structure-function relationship of mitochondria after they are removed from the cell. Despite narratives that first appeared in the classic literature, more recent studies indicate that standard isolation methods actually perturb mitochondrial physiology (DOI: 10.1371/journal.pone.0018317). This makes sense insofar as isolating mitochondria yields a fragmented population, which intrinsically affects the structural aspects of the organelle. This does not even begin to address the perturbations that inevitably arise from losing contact with the cytoskeleton, the nucleus, and other organelles (e.g., the ER), which are increasingly understood to communicate with mitochondria and play essential roles in their normal function. Thus, while AFM may provide a useful orthogonal approach, it must not be portrayed as a technique that neatly approximates bona fide mitochondrial physiology within intact cells. One need go no further than considering that the "height" metric here reflects a relatively spherical organelle. In the cell, normal, functional mitochondria tend to be filamentous and highly networked. Hence, the relevance of this metric is of questionable value if offered as a proxy for mitochondrial morphology measurements in situ. It is necessary to discuss these limitations to provide more context.

Minor comments:

There is a number of places that either lack citations altogether or could benefit from additional citations, which would help readers reliably track the sources of certain claims or statements.

Reviewer #2 (Comments to the Authors (Required)):

The article titled 'Non-Invasive Mechanical-Functional Analysis of Individual Liver Mitochondria by Atomic Force Microscopy' discusses how the mechanical properties of mitochondria as response to various drugs, like CCCP and ADP or rotenon and antimycin A.

The method described is very interesting. Using an AFM probe in tapping mode as a fluctuation sensor is very promising. The data provided proves the correlation in one direction (from cause to effect), however, backwards, deriving states from AFM data on unknown organelles remains an open question.

The work is very interesting and I recommend it for publication after some minor issues are addressed.

Further comments

Setting a low I gain in the feedback of the AFM also introduces a stronger low-pass filter in the feedback loop. How did the Authors identify that the provided setting has a 0 - 500 Hz bandwidth? The JPK AFM software offers some related information, but the Nanoscope software for the Multimode AFM typically does not specify the bandwidth.

In Figure 2.G, the N/T and Succ+Rot data looks to be too close to be four star significant in difference. Please double check. Also in Figure 2 the three N/T data sets in panels C, E and G are quite different. What is the reason behind this, and if they represent the same state, then the overall error would be quite larger than those presented in the panels.

A similar problem is present between Figures 4 and 5, where the Young's modulus is compared with different conditions, but the wild type data is also different between the panels. Notable, in Figure 4.D it is above 20 kPa, in Figure 5.A it is around/below 20 kPa. (At least in Figure 5 A and E they look similar.)

Based on the linear and logarithmic plots of the power spectra, the fitted alpha values have surprisingly low noise assigned to them. It is also somewhat surprising that in Figure S2.B the mica substrate has a broader noise range than the rotenone treated sample in S2.A.

The slope in Figure S2.B should be around 1 for the log-log plot according to the text, but the drawn line changes about 1.0 - 1.5 on a X-range of 2.5, which is a lower value. Please check the Figure.

From Figure S.3 it is apparent that the Young's modulus calculations were performed using a tip radius of about 30 nm and indentation depth of similar value. This may invalidate the model, but would still allow to assume the spherical shape of the tip. Please clarify what the statement 'Fits were restricted to ... < ~20% of the local height' (page 20, line 504). How can one know the local height before fitting the indentation and identifying the contact point in some way?

Reviewer #3 (Comments to the Authors (Required)):

The authors have substantially improved the manuscript. The Introduction is now clear, well structured, and provides a convincing rationale for why a tool as AFM is needed in the field. The authors have appropriately addressed all of the reviewer's previous comments.

Only minor points remain. First, in the Introduction, the justification for the need for careful calibration of fluorescent probes such as TMRM and JC-1 is stated twice (sentences 86 and 99); the authors may consider removing or merging one of these statements to avoid redundancy. Second, in Supplementary Figure S1A, image A does not appear to be visible in the PDF and should be checked and corrected.

1. Thank you for providing a statement on ethics compliance for animal work. Please provide the name of the institution approving the ethics protocols.

Answer: Changed: «All animal work was conducted in accordance with European guidelines for the care and use of laboratory animals (Directive 2010/63/EU) and under protocols approved by the institutional animal ethics committee of the Weizmann Institute of Science»

2. -Please transfer the HIM buffer composition to the section on reagents and solutions. Please cite this section when referring to a reagent/solution for the first time in your methods and do provide a pH for the buffers.

Answer: We moved the full HIM buffer composition to the 'Reagents and Solutions' section and now cite this section at the first mention of HIM/MAS in the Methods; pH values are provided for all buffers.

3. -When acknowledging the use of BioRender for the preparation of figures, please refer to the specific image panels, and if applicable, kindly indicate that BioRender utilises an integrated AI-assisted design model.

Answer: We revised the "Figure preparation" section to specify the exact figure panels assembled in BioRender (Fig. 1E, 1G-K; Fig. 2C-H; Fig. 3A-E; Fig. 4A-E; Fig. 5A-H; Fig. S1F, S1G, S1I, S1K-L). We also added a note that BioRender includes integrated AI-assisted design tools.

4. In the figure legends, please define N/T.

Answer: We revised the figure legends to define N/T at first mention as no treatment (untreated control; no additions).

5. Please provide details on MEF source and preparation. Please also include details on preparation of MTCH2 KO, MFN1 KO or MFN2 KO or provide a relevant citation for each.

Answer: We clarified that the MEF lines were generated and maintained in the laboratory of Atan Gross at the Weizmann Institute of Science and added citations describing the generation of each KO line.

6. -Kindly specify the samples for the section on 'Fixed/dried reference imaging'.

Answer: We clarified the "Fixed/dried reference imaging" section to specify that the fixed/air-dried reference AFM images (Fig. S1D,E) were acquired from WT mouse liver mitochondria (MLM; N/T/no-additions control) adhered to PLL-mica, and that these preparations were used only for morphology/identity controls and not for fluctuation analyses.

7. -Please remove the section summarising availability of supplemental information (lines 612-622) and kindly ensure that all required information from this section is included in the provided table titles or figure titles or figure legends.

Answer: We removed the "Supporting Information Available" section from the main manuscript. Information about the supplementary content is now conveyed via the corresponding figure/table titles and legends. In addition, we added a statement in the figure legends indicating that the per-mitochondrion numerical values underlying all plots are provided in the accompanying Source Data file.

8. Please upload all figure files as individual ones, including the supplementary figure files; all figure legends should only appear in the main manuscript file, and remove the Supporting Information file.
9. Please add a Running Title and a Summary Blurb/Alternate Abstract in our system.
10. Please add a Category for your manuscript in our system.
11. Please add the X and Bluesky handles of your host institute/organisation, as well as your own and/or one of the authors, in our system.
12. Abstract should be a single paragraph not exceeding 175 words.

Answer: Changed

13. Please incorporate any points from the Conclusion section into the Discussion; we only allow a Discussion section.

Answer: Changed

14. Please add your main, supplementary figure, and table legends to the main manuscript text after the references section.

Answer: Changed

15. The "Data Availability" section should be placed after the Materials & Methods section. Please consult our guidelines at <https://www.life-science-alliance.org/manuscript-prep#format>
16. -Please add an Author Contributions section to your main manuscript text and the system.

Answer: Added.

17. Please add a Conflict of Interest statement to your main manuscript text.

Answer: Added.

18. Please be sure that the authorship listing and order is correct.

FINAL FILES:

To upload the final version of your manuscript, please log in to your account:
<https://lsa.msubmit.net/cgi-bin/main.plex>

The following items are required for acceptance.

Thank you for your attention to these final processing requirements. Please revise and format the manuscript and upload materials as soon as you are able.

Thank you for this interesting contribution to literature. We look forward to publishing your paper in Life Science Alliance.

We would like to thank the editor and the reviewers for pertinent and helpful comments. We believe that we have provided accurate and full responses to the comments and hope that the manuscript now meets the standards expected.

Reviewer #1 (Comments to the Authors (Required)):

Zorikova et al. use atomic force microscopy (AFM) to endeavor to correlate different topographical or physical properties of mitochondria with specific functional readouts. They provide genetic and pharmacological evidence that depolarized mitochondria have a lower integrated PSD, whereas increased polarization leads to higher integrated PSD. The relationship between mitochondrial membrane potential and other AFM readouts, e.g., height and Young's modulus, appear inconsistent, however. The authors propose that AFM could provide a method for obtaining mitochondrial structure-function information without the need for dyes or other fluorescent labels. Although the authors provide evidence of a correlation between $\Delta\Psi_m$ and integrated PSD, specifically, the usefulness, and indeed the physiological relevance, of their approach comes with important caveats, which would need to be addressed prior to publication.

We thank the reviewer for the positive and critical comments and hope that the answers and accompanying textual modifications are satisfactory.

Major comments:

1. Lines 42-44: The authors state, "Under various respiratory manipulations, a simple scalar of low frequency height fluctuations closely paralleled the known signal to be unrelated to matrix swelling." This sentence is quite unclear. It is strongly recommended to revise it. For instance, "the known signal to be unrelated" requires clarity and context.

Answer: Corrected to "Across respiratory activators, inhibitors, and uncouplers, the integrated 0-20 Hz fluctuation power correlates with mitochondrial membrane potential ($\Delta\Psi_m$) and does not co-vary with changes in mitochondrial height (a proxy for swelling) under our conditions."

2. Lines 119-120: "The height measurement can serve as a proxy for volume/osmotic state and cristae packing." This sentence has no citation. There appears to be no evidence that height measurement corresponds to "cristae packing." Whether a "tall" mitochondrion has more or fewer or the same number of cristae (per cubic micron) as a "short" mitochondrion is unknown. Thus, this sentence should be revised or removed.

Answer: link to cristae packing was removed

3. Lines 146-149: As a proof that the mouse liver mitochondria (MLM) are purified and functional, the authors provide confocal images of isolated MLM stained with the potentiometric dye TMRM. To demonstrate that these mitochondria harbor a membrane potential, they add the uncoupler CCCP, which should dissipate the TMRM due to depolarization. Figures S1A and S1B require clarification. Firstly, can the authors confirm that a window/level function was applied to S1A and S1B such that the contrast is identical between the two representative images? The signal in S1A is so faint that it is difficult to discern the difference between it and S1B without zooming in a lot. Notably, the brightest puncta, which are presumably isolated organelles, appear to be as large as 5 or 6 microns in diameter. The EM image in S1C, by comparison, shows a field of isolated mitochondria with diameters ranging from 0.5 to 1 micron, as expected. Although the images in S1A and S1B are obtained with confocal microscopy and S1C was taken with EM, the difference in size between the isolated organelles in these images should not be so different (e.g., as much as a factor of 10). The diffraction limit associated with confocal microscopy (on the X-Y axis) is around 200 to 250 nm, thus the considerable size of many of the bright puncta in S1A cannot be due to limited resolution. How can the authors explain this discrepancy? Why show a huge field of view with a 50-micron scale bar? This way of displaying this important proof is not actually providing persuasive evidence. At a minimum, there should be zoomed-in images that clearly highlight the differences which the authors argue exist.

Answer: We agree that the presentation of Fig. S1A-B required clarification. First, we confirm that the representative confocal images in Fig. S1A and S1B were obtained within the same experiment using identical acquisition settings. In addition, in the revised Supporting Information we explicitly state that both panels are displayed using identical linear intensity scaling (same LUT/min-max), so that the contrast is directly comparable between the two images.

Second, the apparent size of the bright punctate objects in these overview images should not be interpreted as the diameter of individual mitochondria. These panels are intended solely as low-magnification (overview) images illustrating the distribution of the TMRM signal across the field of view (50- μ m scale bar). At this field of view and after export/downsampling, submicron mitochondria are visualized as PSF-/sampling-limited points; moreover, some of the brighter objects may correspond to several closely apposed mitochondria (small aggregates) rather than single organelles. Importantly, mitochondrial size and ultrastructure were independently verified by TEM (Fig. S1C) and AFM topography (Fig. S1D-E), which show the expected submicron dimensions.

Finally, to quantitatively demonstrate the presence of $\Delta\Psi_m$ and its collapse upon uncoupling, we emphasize the orthogonal quantitative approaches already presented in Fig. S1: imaging flow cytometry of TMRM (Fig. S1L; identical laser/detector settings within each run and exclusion of debris/aggregates by brightfield-based gating) and ratiometric JC-1 measurements (Fig. S1K), as well as the expected OCR changes in Seahorse assays (Fig. S1F-G).

4. Lines 153-155: "High-resolution AFM of chemically fixed/dried mitochondria provided surface topography for morphology verification (Fig. S1D, E). These images give higher resolution, revealing the cristae folds." On the contrary: these images do not provide evidence of "cristae folds." As the authors will note from S1C, cristae are infoldings of the IMM; the images in S1D and S1E obviously do not capture these ultrastructural details. Thus, this language should be removed.

Answer: Fixed: «High-resolution AFM of chemically fixed and dried mitochondria provided surface topography to verify overall morphology and sample integrity (Fig. S1D,E).»

5. Lines 194-196: "These results demonstrate that AFM-based assessment of integrated power provides a reliable and comparable measure of $\Delta\Psi_m$, on par with established fluorescence-based techniques." The authors must note here that AFM-based assessment of $\Delta\Psi_m$ is indirect and is, by its very nature, correlative. For this reason, it cannot possibly be "on par" with imaging approaches that directly measure the relative proportions of Nernstian dyes on either side of a membrane, which can, in turn, be used to compute absolute voltages of whole organelles or even individual suborganellar membrane potentials. The authors need to moderate this language so as to not overstate the power of their approach.

Answer: Fixed: «These results show that the AFM-derived integrated low-frequency power correlates with $\Delta\Psi_m$ across different metabolic states, providing a label-free, indirect readout that complements established fluorescence-based measurements.»

6. Lines 203-205: "By imaging a field of view containing multiple mitochondria, as shown in Figure 1A, AFM allows one to measure height with nanometer precision." Again, the authors appear to overstate the power of this approach. The mitochondria in Figure 1A, one or two of which is about a micron in diameter, show no signs of submicron detail, which would, of course, correspond to the existence of cristae, which are evident in either EM or super-resolution imaging approaches. The authors must be careful to refrain from making claims that are self-evidently not supported by their data.

Answer: We note that AFM provides images of sub-nm z resolution and several nm xy resolution. This is observed in the dried mitochondria image of figure S1. Our intent was to point out the power of AFM in resolving power. However, the image in Fig. 1A was not intended to show the full resolving power, but to provide a typical image containing several features. We mention height in this context as that is one of the properties used in our data analysis. To emphasize this, we change the sentence to: «When imaging fields of view containing multiple mitochondria (Fig. 1A), AFM can precisely locate individual mitochondria and measure the surface height with nanometer-scale precision in the vertical (Z) dimension.»

7. Lines 232-233: "It is known from transmission electron microscopy experiments that an increase in $\Delta\Psi_m$ can cause swelling of the mitochondrial matrix." This sentence is missing a citation. Furthermore, the construction of this sentence

should be improved. EM cannot provide a readout of $\Delta\Psi_m$, because the mitochondria are fixed or frozen. The classic Hackenbrock studies provide correlative information about the ultrastructure of "condensed" or "orthodox" mitochondria and their bioenergetics. But this is correlative. This fact should come across more clearly in the language here. This statement is further complicated by the fact that super-resolution imaging shows that depolarization of mitochondria with the uncoupler FCCP induces dramatic swelling of the mitochondrial matrix. Thus, while the authors state that it is known that an increase $\Delta\Psi_m$ causes swelling, it is also evident that a diminished $\Delta\Psi_m$ occurs upon depolarization. Of course, this complicates the picture substantially.

Answer: We thank the reviewer for this important clarification. We have revised the statement to avoid implying that EM directly measures $\Delta\Psi_m$ and to explicitly frame the evidence as correlative. Specifically, we replaced the original sentence with the following text:

“Ultrastructural studies have reported associations between mitochondrial energetic state and matrix configuration/volume; however, EM provides static, correlative snapshots and does not directly quantify $\Delta\Psi_m$, and swelling responses are condition dependent. Moreover, increases in mitochondrial volume have also been reported upon depolarization/uncoupling.”

We also added the appropriate references (including the classical Hackenbrock studies and reports describing mitochondrial volume increases upon uncoupling/depolarization), thereby addressing the lack of citations and the condition-dependent nature of matrix swelling.

8. Lines 234-263: The authors go on for some time about effects of valinomycin and $\Delta\Psi_m$. While they indicate that valinomycin is known to dissipate $\Delta\Psi_m$ in literature, whether valinomycin has the same effect in their hands is unclear—namely, in Figure 3D, the violin plot purports to show a significant change in $\Delta\Psi_m$ between "NT" and "Val," but it is not clear if the change is up or down. Indeed, the nature of the violin plot makes it extremely difficult, at least in this case, to see where the mean would lie, exactly. Please indicate what the meaning is numerically in either condition to clarify whether the difference is an increase or decrease upon valinomycin treatment. Concordantly, the data in Figure 3E indicate that there is no change in $\Delta\Psi_m$. Thus, the overall conclusions drawn as to the effects of valinomycin on the $\Delta\Psi_m$ and the relationship to the AFM parameters in Figures 3A-3C are ambiguous. By contrast, CCCP has a clear effect on membrane potential in both the TMRM- and JC-1-stained mitochondria, so any conclusion about the relationship between $\Delta\Psi_m$ and the correlated AFM parameters would be more tenable with respect to this treatment. The authors' statement that "These structural and mechanical changes were accompanied by a loss of $\Delta\Psi_m$, confirmed by reduced TMRM mean pixel intensity (Fig. 3D) and lower JC-1 red/green ratio (Fig. 3E)." is not actually borne out by their data—particularly in the case of Fig. 3E, which shows no significant difference. This data must be revisited, as the specific conclusions appear to be unwarranted. If

the authors wish to keep the valinomycin data, it would be recommended to perform a dose-response to determine a concentration that induces clear depolarization (shown by both loss of TMRM and a decrease in the red/green ratio of JC-1; conversely, this ambiguous data could be removed, as it does not appear to add much value to the overall picture).

Answer: We agree that the relationship between integrated low-frequency fluctuation power and mitochondrial swelling was not clear. We therefore revised this section to (i) frame the evidence from ultrastructural studies as correlative rather than implying that EM directly reports $\Delta\Psi_m$, and (ii) explicitly test whether integrated power could be explained by matrix swelling/volume changes.

Specifically, we updated the text to note that reported links between energetic state and matrix configuration/volume are associations based on static EM snapshots and that swelling responses are condition-dependent, with matrix volume dynamically responding to ion fluxes and $\Delta\Psi_m$. We also clarified the experimental logic for using valinomycin as an osmotic-swelling perturbation: valinomycin promotes matrix K^+ influx with osmotically coupled water entry and can partially dissipate $\Delta\Psi_m$ via the K^+ cycle; because valinomycin was applied after pre-energizing with malate/glutamate, its effects are interpreted primarily relative to the Mal+Glut condition (the immediate pre-treatment state), with NT serving as a baseline reference.

Using this approach, we show that valinomycin causes an increase in apparent height (consistent with osmotic swelling) and a decrease in apparent Young's modulus (softening) (Fig. 3B,C), while integrated power decreases (Fig. 3A). Importantly, this directionality (height increasing while integrated power decreases) is opposite to the co-directional changes observed across respiratory manipulations in Figures 1-2, indicating that integrated power does not simply mirror swelling-driven height changes and is not a proxy for swelling. Consistently, Pearson correlations across the Fig. 3 conditions show that neither height nor Young's modulus strongly correlates with integrated power (all $|r| < 0.3$).

Finally, we moderated our interpretation of $\Delta\Psi_m$ under valinomycin: CCCP produces a robust $\Delta\Psi_m$ loss in both TMRM and JC-1 readouts (Fig. 3D,E), whereas the valinomycin effect is more modest and assay-dependent (JC-1 does not show a significant NT vs Val change under our conditions). Therefore, we interpret valinomycin-associated AFM changes without inferring a definitive $\Delta\Psi_m$ collapse and rely on CCCP as the clearest perturbation linking $\Delta\Psi_m$ loss to the correlated AFM readouts (Fig. 3D,E; Table S1 for group means and statistics).

Also, we added the following sentence to the manuscript text: "In contrast, valinomycin produced a more modest and assay-dependent change in $\Delta\Psi_m$: TMRM decreased relative to Mal+Glut and relative to N/T (one-way ANOVA with Tukey's multiple comparisons; Val vs Mal+Glut, $p \leq 0.0001$; Val vs N/T, $p \leq 0.0001$), whereas the JC-1 red/green ratio did not significantly differ between N/T and Val (Tukey, $p = 0.7567$) (Fig. 3D,E)."

9. Lines 1, 41, 287-289, 308, 407, etc.: References to "nanomotions" and "nanomechanics" and "nanoscale" should be removed, as the provided AFM images do not reflect the measurement of nm-resolution structures in this study.

Answer: The main findings of this article with respect to the AFM measurements involved the nm-level fluctuations recorded for the mitochondria under different environments. Furthermore, the mechanical measurements recorded nanoNewton forces while indenting the sample by a few tens of nm. Furthermore, the AFM capability to resolve nm-level modulations on the surface morphology as shown in the dried species image in Fig. S1 (and demonstrated below with a cross-section profile explicitly showing the nm-level fluctuations) was only referred to here in passing but could be utilized in further work to further elucidate the structural properties of mitochondria. Therefore, our usage is in accordance with the scale of measurements and common literature usage. We understand the referee's concern that we do not provide examples or use information on xy (in sample plane) resolution at the nm scale, but for reasons mentioned here do believe that the nomenclature is appropriate.

10. Figures 3-5 are all missing representative images of the AFM. It is difficult to assess the effects of pharmacological and genetic perturbations without being able to examine the actual images themselves. The authors must include representative images in these figures.

Answer: We understand the reviewer's request for representative AFM images. In our study, however, Figs. 3-5 are not AFM imaging figures; they summarize quantitative AFM-derived parameters at the single-organelle level (integrated 0-20 Hz fluctuation power, AFM height, and apparent Young's modulus) as distributions across mitochondria and biological replicates. Although we could selectively find "representative images" the results here are from statistics on large number of measurements on several biological repeats. Such images would not capture the variability that is central to our conclusions. Since the figures are already quite busy, we

focus these figures on the quantitative readouts themselves which give pertinent information.

Representative AFM topography and stiffness maps used for organelle identification/validation are already shown in Fig. 1A and Fig. 2A-B, and additional AFM topography identity controls are provided in Fig. S1D-E; AFM topography is explicitly used to guide probe placement as described in the Methods and to measure height but changes in the distinct morphologies are not central to our study.

11. Lines 343-360: The discussion goes into some detail about how this study showed new information about MFN2 and the coenzyme Q10 pool. Which figures or panels reflect this narrative?

Answer: The MFN2-related narrative is reflected in Fig. 5E-H. Specifically, MFN2 KO mitochondria exhibit increased stiffness (Fig. 5E), no significant height changes relative to WT (Fig. 5F) and decreased integrated power and JC-1 red/green ratios (Fig. 5G-H), indicating a distinct mechanobioenergetic phenotype. We revised the Discussion to explicitly reference these panels and to avoid implying that we directly measured the coenzyme Q pool; CoQ/CoQ10 is now discussed only as mechanistic context from prior literature.

12. Lines 368-372: As indicated above, the valinomycin data is inconclusive.

Answer: As detailed in our response to comment 8, we agree that under our experimental conditions valinomycin does not produce a uniformly strong $\Delta\Psi_m$ collapse across both fluorescent assays, and we have therefore revised the manuscript to avoid drawing a definitive depolarization conclusion from valinomycin. Instead, we now describe the valinomycin effect on $\Delta\Psi_m$ as modest and assay-dependent and report it numerically with statistics: TMRM decreases under valinomycin (NT: 21.151 ± 0.354 a.u.; Val: 18.081 ± 0.193 a.u.; one-way ANOVA with Tukey, $p \leq 0.0001$), whereas the JC-1 red/green ratio does not significantly differ between NT and Val (NT: 3.240 ± 0.131 ; Val: 2.767 ± 0.067 ; Tukey, $p = 0.7567$) (Fig. 3D,E; Table S1).

Importantly, we do not consider the valinomycin dataset inconclusive with respect to the main purpose for which it is used in Fig. 3. As clarified in the revised text (comment 8 response), valinomycin is included primarily as a K^+ /osmotic perturbation: it yields clear AFM phenotypes consistent with swelling/softening (height increases while Young's modulus decreases; Fig. 3B,C), while integrated low-frequency fluctuation power decreases (Fig. 3A). This opposite directionality (height increase with integrated power decrease) relative to the co-directional changes observed across respiratory manipulations in Figs. 1-2 demonstrates that integrated power is not a proxy for swelling-driven height changes.

Accordingly, our inference linking $\Delta\Psi_m$ loss to the correlated AFM readouts is based on CCCP, which produces a robust decrease in both TMRM and JC-1 readouts, whereas valinomycin-associated AFM changes are interpreted without inferring a definitive $\Delta\Psi_m$ collapse.

13. Given the emphasis of this study on examining the topography of mitochondria using AFM on isolated mitochondria, it would be appropriate to discuss the limitations of investigating the structure-function relationship of mitochondria after they are removed from the cell. Despite narratives that first appeared in classic literature, more recent studies indicate that standard isolation methods perturb mitochondrial physiology (DOI: 10.1371/journal.pone.0018317). This makes sense insofar as isolating mitochondria yields a fragmented population, which intrinsically affects the structural aspects of the organelle. This does not even begin to address the perturbations that inevitably arise from losing contact with the cytoskeleton, the nucleus, and other organelles (e.g., the ER), which are increasingly understood to communicate with mitochondria and play essential roles in their normal function. Thus, while AFM may provide a useful orthogonal approach, it must not be portrayed as a technique that neatly approximates bona fide mitochondrial physiology within intact cells. One need go no further than considering that the "height" metric here reflects a relatively spherical organelle. In the cell, normal, functional mitochondria tend to be filamentous and highly networked. Hence, the relevance of this metric is of questionable value if offered as a proxy for mitochondrial morphology measurements in situ. It is necessary to discuss these limitations to provide more context.

Answer: We added the limitations into the discussion section « All AFM measurements in this study are performed on isolated mitochondria adhered to a solid support under ex vivo conditions. Isolation and handling can perturb mitochondrial physiology and enrich for fragmented, rounded organelles, and removal from the cellular environment eliminates interactions with the cytoskeleton and with other organelles (including ER contact sites) that shape mitochondrial behavior in situ⁴⁶. Accordingly, the AFM-derived parameters reported here should be interpreted as comparative ex vivo single-organelle phenotypes rather than a direct representation of mitochondrial physiology within intact cells. In particular, "height" in our assay reflects organelle geometry after isolation and adhesion and is not intended as a proxy for filamentous/network morphology in living cells.»

Minor comments:

There is a number of places that either lack citations altogether or could benefit from additional citations, which would help readers reliably track the sources of certain claims or statements.

Added:

30 — Hackenbrock CR (1966) Ultrastructural bases... I. 10.1083/jcb.30.2.269

32 — Javadov S et al. (2018) Modeling analysis of mitochondrial swelling. 10.1016/j.mito.2017.08.004

33 — Kaasik A et al. (2007) Regulation of mitochondrial matrix volume. 10.1152/ajpcell.00272.2006

- 34 — Safiulina D et al. (2006) $\Delta\Psi$ loss \leftrightarrow increase in mitochondrial volume. 10.1002/jcp.20476
- 35 — Gerencser AA et al. (2008) Mitochondrial swelling measurement in situ... 10.1529/biophysj.107.118620
- 36 — Garlid KD, Paucek P (2003) Mitochondrial potassium transport: the K(+) cycle. 10.1016/S0005-2728(03)00108-7
- 45 — Radmacher M et al. (1994) Direct observation of enzyme activity with AFM. 10.1126/science.8079171
- 46 — Picard M et al. (2011) Isolation methods disrupt mitochondrial structure/function. 10.1371/journal.pone.0018317
- 50 — Goldberger AL et al. (2002) Fractal dynamics in physiology... 10.1073/pnas.012579499
- 52 — Stupar P et al. (2017) Mitochondrial activity detected by cantilever sensor. 10.5194/ms-8-23-2017
- 53 — Vyshenskaya TV et al. (2014) Dynamic phase microscopy... ER oscillations. 10.1134/S0006297914090077
- 56 — Walewska A et al. (2022) Measuring mitochondrial potassium channels: critical assessment. 10.3390/ijms23031210
- 57 — Mourier A et al. (2015) Mfn2 required for CoQ levels. 10.1083/jcb.201411100
- 58 — Sebastián D et al. (2012) Mfn2 links mitochondria–ER + insulin signaling. 10.1073/pnas.1108220109
- 59 — Ding Y et al. (2015) Mfn2-deficiency & autophagy/proliferation. 10.1371/journal.pone.0121328
- 60 — de Brito OM, Scorrano L (2009) Mfn2 regulates mito–ER tethering (Ras). 10.1016/j.mito.2009.02.005
- 61 — Filadi R et al. (2015) Mfn2 ablation increases ER–mitochondria coupling. 10.1073/pnas.1504880112

Reviewer #2 (Comments to the Authors (Required)):

The article titled 'Non-Invasive Mechanical-Functional Analysis of Individual Liver Mitochondria by Atomic Force Microscopy' discusses how the mechanical properties of mitochondria as response to various drugs, like CCCP and ADP or rotenon and antimycin A.

The method described is very interesting. Using an AFM probe in tapping mode as a fluctuation sensor is very promising.

The data provided proves the correlation in one direction (from cause to effect), however, backwards, deriving states from AFM data on unknown organelles remains an open question.

The work is very interesting, and I recommend it for publication after some minor issues are addressed.

Further comments

1. Setting a low I gain in the feedback of the AFM also introduces a stronger low-pass filter in the feedback loop. How did the Authors identify that the provided setting has a 0 - 500 Hz bandwidth? The JPK AFM software offers some related information, but the Nanoscope software for the Multimode AFM typically does not specify the bandwidth.

Answer: The bandwidth was measured by scanning over a small (25 nm) step on a calibration grid, at various scanning speeds, under conditions used in the experiments. The height signal was then recorded on an external scope thus giving the time-reponse of the system after feedback (low-pass filter). The rise time (time required to reach 67% of maximal value) reached 2 ms at the fastest scanning speeds which corresponds to 500 Hz. It should be pointed out that this is a maximum response time since it includes also the time required to scan over the step. Since this time is much faster than that needed to accurately measure the frequencies recorded in the noise measurements, it was considered sufficient to show that our measurements are not limited by system response. Experimental section now summarizes this

2. In Figure 2.G, the N/T and Succ+Rot data looks to be too close to be four star significant in difference. Please double check.

Answer: We re-checked the statistics for Fig. 2G (mitochondrial height, Complex II-linked conditions). Although the distributions partially overlap visually, the group means are clearly separated: N/T = 334.4 ± 8.2 nm vs. Succ+Rot = 398.0 ± 10.0 nm ($n = 30$ mitochondria per condition, from $N = 3$ biological replicates). One-way ANOVA is significant ($F(3,116) = 63.85$, $p = 1.89 \times 10^{-24}$), and Tukey's multiple-comparisons test confirms that the N/T vs. Succ+Rot difference remains highly significant ($p \leq 0.0001$; denoted as ****). We have clarified this in the revised manuscript and included the full statistical summary in Supplementary Table S1.

3. Also in Figure 2 the three N/T data sets in panels C, E and G are quite different. What is the reason behind this, and if they represent the same state, then the overall error would be quite larger than those presented in the panels.

Answer: We agree that the N/T values differ across panels C, E, and G. This is because these panels represent three independent experimental condition sets

acquired in separate sessions/mitochondrial preparations, each with its own contemporaneous N/T control measured on the same day and preparation as the other conditions in that set. Specifically: (C/D) N/T-Mal+Glu-Rot-CCCP; (E/F) N/T-TMPD+Asc-ADP-Azide; (G/H) N/T-Succ+Rot-ADP-Antimycin A.

Absolute height values can vary between preparations/days (preparation-to-preparation/batch variability), therefore N/T distributions are not intended to be compared across panels. The error shown in each panel reflects within-set variability of the matched control and conditions acquired in that same session. Accordingly, all statistical tests are performed within each panel/set relative to its matched N/T control. We clarified this in the legend of Fig. 2. «WT data shown in this figure are experiment-matched controls acquired in parallel with the indicated condition(s) during the same measurement session; therefore, WT values should not be compared across different figures/experiments. Statistical comparisons are performed within each figure against its matched WT control.»

4. A similar problem is present between Figures 4 and 5, where the Young's modulus is compared with different conditions, but the wild type data is also different between the panels. Notable, in Figure 4.D it is above 20 kPa, in Figure 5.A it is around/below 20 kPa. (At least in Figure 5 A and E they look similar.)

Answer: We agree that the absolute WT Young's modulus values differ between Fig. 4D and Fig. 5A. This is because Figures 4 and 5 correspond to independent experimental datasets (different genetic models and measurement sessions/cell preparations). In each figure, the WT group represents the matched, contemporaneous control measured in parallel with the corresponding condition(s) (same session, same culture batch/passage and instrument settings). Absolute AFM-derived stiffness values can vary between sessions due to day-to-day / culture-to-culture variability (e.g., passage, confluency, cell spreading) and other session-dependent factors; therefore, WT values across different figures are not intended for direct comparison. The error bars shown reflect within-experiment variability, and all statistical comparisons are performed within each figure against its matched WT control. We have clarified this explicitly in the figure legends to avoid cross-figure interpretation. «WT data shown in this figure are experiment-matched controls acquired in parallel with the indicated condition(s) during the same measurement session; therefore, WT values should not be compared across different figures/experiments. Statistical comparisons are performed within each figure against its matched WT control. »

5. Based on the linear and logarithmic plots of the power spectra, the fitted alpha values have surprisingly low noise assigned to them. It is also somewhat surprising that in Figure S2.B the mica substrate has a broader noise range than the rotenone treated sample in S2.A.
6. Answer: The standard deviation was computed using Excel "LINEST" function which performs an ordinary least squares computation on the data. This presumes no errors in x values which holds in our study where x axis is the frequency. This returns the slope, intercept, standard deviation of both, R², standard deviation of y, Fisher F statistic, degrees of freedom, regression SSR,

and errors SSE We attach a sample spreadsheet with calculation for example (for Rotenone-treated sample). We report the standard deviation (σ) as obtained from this analysis. We would like to thank the reviewer for noticing that Figure S2A does not seem to correspond to the expected power amplitudes. This was due to an error in figure preparation. The correct graphs have now been inserted. The analysis originally reported was done on the correct data so slope and uncertainty values remain unchanged (the complete data and analysis seen in the provided sample excel file). As for the noise range (std. deviation) this is a function of the measurement, and not of the fluctuation noise itself which is the parameter used for comparison and given as slope α . The slope in Figure S2.B should be around 1 for the log-log plot according to the text, but the drawn line changes about 1.0 - 1.5 on a X-range of 2.5, which is a lower value. Please check the Figure.

Answer: The text mentions that slope for mica is around 0.3 which corresponds to the line shown in the graph. Value of 1 is expected for a viable organism as in S2-C.

7. From Figure S.3 it is apparent that the Young's modulus calculations were performed using a tip radius of about 30 nm and indentation depth of similar value. This may invalidate the model but would still allow to assume the spherical shape of the tip. Please clarify what the statement 'Fits were restricted to ... < ~20% of the local height' (page 20, line 504). How can one know the local height before fitting the indentation and identifying the contact point in some way?

Answer: From Figure S3 we can see that the total movement of z-piezo is about 30 nm after contact with surface. The total bend of cantilever over this distance is about 100 pN which corresponds to cantilever bend of about 1 nm so that the maximal penetration is also about 30 nm. The mitochondria height ranged between 200 - 400 nm. The fit is done post-experiment. The height is thus available when the fit is performed. Since in the QI mode we obtain topography as well as force-curves at each pixel, on each experiment we can choose conditions giving appropriate depths based on viewing a few preliminary force curves. We now include this fact in the experimental section. The reviewer is correct that choosing a depth similar to the tip radius does not conform to the assumed condition for Hertz linear elastic model, but it still gives values similar to those found by others for mitochondrial elasticity (J. Phys. Chem. B 2023, 127, 50, 10778-10791) and we are careful to maintain consistent measurement conditions as we are looking for differences between the samples and not precise values.

Reviewer #3 (Comments to the Authors (Required)):

The authors have substantially improved the manuscript. The Introduction is now clear, well structured, and provides a convincing rationale for why a tool as AFM is needed in the field. The authors have appropriately addressed all of the reviewer's previous comments.

1. Only minor points remain. First, in the Introduction, the justification for the need for careful calibration of fluorescent probes such as TMRM and JC-1 is stated twice (sentences 86 and 99); the authors may consider removing or merging one of these statements to avoid redundancy. Second, in Supplementary Figure S1A, image A does not appear to be visible in the PDF and should be checked and corrected.

Answer: Corrected

February 12, 2026

RE: Life Science Alliance Manuscript #LSA-2025-03602R

Prof. Atan Gross
Weizmann Institute of Science
Biological Regulation Department
The Weizmann Institute of Science
Candiotty Bldg, Rm 306a
Rehovot, IL-Rehovot 76100 76100
Israel

Dear Dr. Gross,

Thank you for submitting your Research Article entitled "Bridging Nanomechanics and Bioenergetics of Single Mitochondria by Atomic Force Microscopy". We acknowledge all the revisions made to address the pending concerns of the reviewers and to meet some of our formatting guidelines.

We have discussed your revised manuscript within our editorial team and it is a pleasure to let you know that your manuscript is now accepted for publication in Life Science Alliance. Congratulations on this interesting work!

DISTRIBUTION OF MATERIALS:

Again, congratulations on a very nice paper. I hope you found the review process to be constructive and are pleased with how the manuscript was handled editorially. We look forward to future exciting submissions from your lab.

Sincerely,

Sarita Hebbar, PhD
Scientific Editor
Life Science Alliance
<http://www.lsajournal.org>